

# Global clear-sky surface skin temperature from multiple satellites using a single-channel algorithm with viewing zenith angle correction

Benjamin R. Scarino[1], Patrick Minnis[2], Thad Chee[1], Kristopher M. Bedka[2], Christopher R. Yost[1], and Rabindra Palikonda[1]

[1]Science Systems and Applications, Inc., One Enterprise Pkwy Ste 200, Hampton, VA 23666 USA
[2]NASA Langley Research Center, 21 Langley Blvd MS 420, Hampton, VA 23681-2199 USA

*Correspondence to*: Benjamin R. Scarino (benjamin.r.scarino@nasa.gov)

**Abstract.** Surface skin temperature ($T_s$) is an important parameter for characterizing the energy exchange at the ground/water-atmosphere interface. The Satellite ClOud and Radiation Property retrieval System (SatCORPS) employs a single-channel thermal-infrared- (TIR-) method to retrieve $T_s$ over clear-sky land and ocean surfaces from data taken by geostationary-Earth orbit (GEO) satellite and low-Earth orbit (LEO) satellite imagers. GEO satellites can provide somewhat continuous estimates of $T_s$ over the diurnal cycle in non-polar regions, while polar $T_s$ retrievals from LEO imagers, such as the Advanced Very High Resolution Radiometer (AVHRR) can complement the GEO measurements. The combined global coverage of remotely sensed $T_s$, along with accompanying cloud and surface radiation parameters, produced in near-real time and from historical satellite data, should be beneficial for both weather and climate applications. For example, near-real-time hourly $T_s$ observations can be assimilated in high-temporal resolution numerical weather prediction models and historical observations can be used for validation or assimilation of climate models. Key drawbacks to the utility of TIR-derived $T_s$, data include the limitation to clear-sky conditions, the reliance on a particular set of analyses/reanalyses necessary for atmospheric corrections, and the dependence on viewing angle. Therefore, $T_s$ validation with established references is essential, as is proper evaluation of $T_s$ sensitivity to atmospheric correction source.

This article presents improvements on the NASA Langley GEO satellite and AVHRR TIR-based $T_s$ product, derived using a single-channel technique. The resulting clear-sky skin temperature values are validated with surface references and independent satellite products. Furthermore, an empirical means of correcting for the viewing-angle dependency of satellite land surface temperature (LST) is explained and validated. Application of a daytime nadir-normalization model yields improved accuracy and precision of GOES-13 LST relative to independent Moderate-resolution Imaging Spectroradiometer (MYD11_L2) LST and Atmospheric Radiation Measurement Program/NOAA ESRL Surface Radiation network ground stations. These corrections serve as a basis for a means to improve satellite-based LST accuracy, thereby leading to better monitoring and utilization of the data. The immediate availability and broad coverage of these skin temperature observations should prove valuable to modelers and climate researchers looking for improved forecasts and better understanding of the global climate model.

## 1 Introduction

Surface skin temperature ($T_s$) is a critical quantity for characterizing the exchange of energy between the Earth's surface and the atmosphere. Consistent land and ocean measurements of $T_s$ are essential for regional and global climate assessment and weather model data assimilation. Surface energy balance and top-of-atmosphere (TOA) radiative budget calculations rely on the accuracy of these surface parameters (Bodas-Salcedo et al., 2008). In addition to surface flux analyses, $T_s$ retrievals are used to minimize model prediction uncertainty by updating model state values with observations at regular time steps – an important consideration for climate and numerical weather prediction (NWP) models (Garand, 2003; Tsuang et al., 2008; Reichle et al.,





2010; Ghent et al. 2010; Guillevic et al., 2012; Draper et al., 2014). The modeling community could benefit significantly from the provision of frequent, spatially contiguous, global land and ocean $T_s$ data (Rodel et al. 2004, Bosilovich et al. 2007). Many other uses of $T_s$ as well as the status and future of $T_s$ retrievals are summarized by Li et al. (2013). It is clear that the need is growing for higher accuracy, global coverage, and greater temporal and spatial resolution of $T_s$ retrievals from satellite imager

data.

Satellite-based $T_s$ retrieval, validation, and modeling studies originate from a variety of sources, e.g., the National Environmental Satellite, Data, and Information Service (NESDIS) and the National Oceanic and Atmospheric Administration (NOAA) via the Advanced Very High Resolution Radiometer (AVHRR) series and the Geostationary Operational Environmental Satellite (GOES) sensors (Prata, 1993, 1994; Coll and Caselles, 1997; Sobrino and Raissouni, 2000; Kerr et al., 2004; Sobrino et al.,

2004; Yu et al., 2009, 2010, 2012; Sun et al., 2012). Specifically, using a single-channel land surface temperature (LST) algorithm, Heidinger et al. (2013) found good agreement with ground sources in a verification study of GOES and AVHRR Pathfinder Atmospheres–Extended (PATMOS-x) LST. Furthermore, near-real-time LST is produced operationally from Meteosat Spinning Enhanced Visible and Infrared Imager (SEVIRI) data, which offer continuous coverage of Europe and Africa, and served as the focus of several LST validation studies (DaCamara, 2006; Kabsch et al., 2008; Trigo et al., 2008; Göttsche et

al., 2013). Retrievals using radiances from the Moderate Resolution Imaging Spectroradiometer (MODIS) have been both the target and standard for a number of LST verification studies (Wan et al., 2002, 2004, 2008; Coll et al., 2009; Jiménez et al., 2012). Duan et al. (2014) used four daily observations from Terra- and Aqua-MODIS to capture the diurnal cycle of LST, which is critical for full characterization of the climate system. Wang et al. (2014) conducted a three-way $T_s$ comparison using MODIS, in situ ground observations, and model simulations. They note the high importance of accurate cloud-clearing and the inherit

difficulties of resolution scaling when comparisons are conducted between satellite data and point references – conclusions supported in a similar MODIS daytime LST verification study conducted by Williamson et al. (2013).

With more reliable calibrations, operational GEO and low Earth-orbiting (LEO) satellite imagers are being used to derive cloud and radiation properties in near real time (NRT), e.g., Minnis et al. (2008a). The combination of GOES-East (GOES-13), GOES-West (GOES-15), Meteosat Second Generation (MSG; Meteosat-9 or Meteosat-10), MTSAT-2 (recently replaced by Himawari-

8), and the Indian Space Research Organization INSAT-3D provides high-temporal resolution (1-hour nominal) quasi-global $T_s$ data produced in NRT, with a shared single-channel retrieval algorithm (e.g., Fig. 1). The methodology (Section 2) is flexible and easily transportable to other GEO and LEO imagers, including the current AVHRR instruments on the NOAA and EUMETSAT MetOp platforms. Near-real-time AVHRR $T_s$ retrievals supplement the GEO data and fill in missing measurements over polar regions (e.g., Fig. 2). This same method is being applied to historic and current imager datasets, particularly as part of

the Satellite ClOud and Radiative Property retrieval System (SatCORPS) analyses of AVHRR data for provision of a NOAA Climate Data Record (Minnis et al., 2016), and for MODIS, GEO, and Suomi-National Polar-Orbiting Partnership (S-NPP) Visible Infrared Imaging Radiometer Suite data as part of the Clouds and Earth's Radiant Energy System (CERES) project (e.g., Minnis et al., 2010).

This article highlights recent improvements made to the SatCORPS NRT satellite $T_s$ product (Scarino et al., 2013), via

comparisons of GOES and AVHRR $T_s$ retrievals with established sea surface temperature (SST; Section 4) and LST (Section 5) reference datasets. The influence of NWP source on retrieved $T_s$ values is also examined. The main improvements over the earlier version are enhanced pixel-level resolution output and hourly GEO retrieval time steps. The SatCORPS $T_s$ retrieved from GOES and AVHRR data are evaluated by comparing with reference datasets based on in situ, surface, and satellite measurements. In addition to the validation comparisons, an empirical means of correcting for LST viewing angle dependency is

developed and tested against the reference datasets. The combined GEO and AVHRR retrievals allow for high-resolution



temporal monitoring of the $T_s$ diurnal cycle, an essential state variable for numerical weather model data assimilation and climate studies (e.g., Draper et al., 2014). The $T_s$ products and uncertainties described here should be valuable for improving surface energy flux analyses and numerical weather prediction owing to their NRT global availability over land and ocean.

## 2  Data

### 2.1 Satellite data for surface skin temperature retrieval

Clear-sky surface skin temperature is retrieved from channel 4 (11 μm) radiances taken by the NOAA-18 AVHRR for the period January –December 2008 in the Global Area Coverage (GAC) format. The nominal satellite equatorial crossing time is 13:30 LT during that time period. A GAC pixel radiance is formed by averaging the radiances of four consecutive raw 1-km AVHRR pixels along the scan direction. The process is repeated after skipping the fifth pixel and so on to produce consecutive GAC pixels along the scan line. Two scan lines are then skipped and the pixel averaging is applied again to the third scan line. Thus, a GAC pixel nominally covers a 1-km x 4-km area (a 2 km$^2$ pixel), but because of sampling, represents a 3-km × 5-km area that yields an effective resolution of ~4 km. The AVHRR data were analyzed with the SatCORPS-A1 methodology (Minnis et al., 2016) to retrieve cloud properties, TOA broadband fluxes, and clear-sky surface skin temperature. Clear pixels are determined from the SatCORPS cloud mask. Details of the skin temperature retrieval process are given in Section 3.

Hourly channel-4 (10.8 μm) data from GOES-13 (GOES-East) and GOES-15 (GOES-West) taken during January, April, July, and October (hereafter, JAJO) 2013 are used to retrieve $T_s$ for validation with surface and other satellite surface skin temperature datasets. Furthermore, GOES-13 and GOES-15 data are employed to test the viewing angle correction parameterization. The nominal GOES imager resolution is 4 km. The pixels are sub-sampled, however, to an effective resolution of 8-km during full disk and hourly hemispheric scans. These data were analyzed with a version of SatCORPS-A1 adapted to the GOES channels as described by Minnis et al. (2008a).

Aqua-MODIS data taken over the GOES-East domain were analyzed with the CERES Ed4 retrieval code (Minnis et al., 2008b, 2010, 2011) to match with the GOES-13 LST retrievals during JAJO 2013. The matched data are used to develop a parameterization to correct LST for viewing zenith angle (VZA) dependence. The MODIS data are taken twice per day at a 1-km resolution within 1.5 hours of ~0130 and 1330 LT.

### 2.2 Validation data

For validation comparisons, this study employs surface and satellite-based references. The SatCORPS AVHRR SST values are compared to the daily high-resolution blended SST analysis described by Reynolds et al. (2007). It comprises the NOAA "Optimum Interpolation" SST (OI SST) Version-2 high-resolution dataset, which consists of a global 0.25° × 0.25° grid of blended satellite (AVHRR two- and three-channel algorithms) and in situ measurements of daily SST. It covers the period from January 1981 to the present.

Surface radiometer measurements are used to validate the SatCORPS AVHRR and GOES LST values. Two surface datasets are employed: the NOAA ESRL Surface Radiation (SURFRAD) network upwelling/downwelling Eppley hemispheric Precision Infrared Radiometer (PIR) broadband longwave fluxes (Augustine et al., 2000), and the Atmospheric Radiation Measurement (ARM) Southern Great Plains (SGP) Central Facility (36.3°N, 97.5°W) 11-μm upwelling/downwelling infrared thermometer (IRT) brightness temperatures (Morris, 2006). Both datasets are common references for evaluating LST retrievals over the contiguous United States (Guillevic et al., 2012; Yu et al., 2012; Heidinger et al., 2013; Wang et al., 2014).



The ARM IRT ground-based radiation pyrometers provide measurements of the equivalent blackbody brightness temperature for the 9.6-11.5-μm spectral band every 60 seconds. From a 10-meter-height with 30.5° FOV, the upwelling IRT measures the effective ground radiating temperature, i.e., the temperature equivalent of the ground infrared radiant energy assuming the surface emissivity ($\varepsilon_s$) is equal to 1.0 (Morris 2006). A true skin temperature $T_s$ can, therefore, be determined as

$$T_s = B^{-1}\left\{\frac{\left[B(T_o) - (1 - \varepsilon_s) \times B(T_{o\downarrow})\right]}{\varepsilon_s}\right\},$$

(1)

where $\varepsilon_s$ is from the CERES 11-μm database (e.g., Chen et al., 2004) and the spectral downwelling narrowband brightness temperature ($T_{o\downarrow}$), which is measured by a 2-meter-height up-looking IRT. The Planck function for the particular waveband is $B(T)$, and $T_o$ is temperature equivalent to the surface-leaving blackbody radiance. Note that the ARM downwelling IRT at the Lamont, OK Central Facility was no longer operating in 2013, therefore $T_{o\downarrow}$ was acquired from the nearby Lamont, OK Extended Facility downwelling IRT, which operates in unison with the Central Facility instrument. It is expected that there is negligible variation in $T_{o\downarrow}$ over the ~200-m distance between the two sites.

The SURFRAD network currently consists of seven stations situated in geographically diverse regions across the continental United States. At each station, two PIRs measure the upwelling and downwelling broadband longwave thermal infrared irradiance ($LW_\uparrow$ and $LW_\downarrow$) in the spectral range from 3.0 to 50.0 μm every 60 seconds (every 180 seconds in 2008). Surface skin temperature is determined from $LW_\uparrow$ and $LW_\downarrow$ by

$$T_s = \left[\frac{LW_\uparrow - (1 - \varepsilon_B) \times LW_\downarrow}{\varepsilon_B \times \sigma}\right]^{1/4},$$

(2)

where $\varepsilon_B$ is the CERES broadband emissivity (Wilber et al., 1999) and $\sigma$ is the Stefan-Boltzmann constant.

Another LST reference dataset used here is the Version-5 Aqua-MODIS LST/Emissivity product (MYD11_L2; hereafter, MYD11), which is derived from the generalized split-window algorithm (Wan and Dozier 1996; Wan and Li, 1997; Snyder and Wan, 1998). It includes values of LST and surface spectral emissivity values retrieved from clear-sky 1-km MODIS pixels. Because MYD11 is derived from different data using a different type of algorithm, and is accurate to ±1 K or less (Wan et al., 2002, 2004; Wan, 2008), it serves well as an independent reference for comparing with the GOES retrievals.

### 2.3 Reanalysis input

Model data are used as input to compute TOA brightness temperatures ($T_{toa}$). These include the model surface air ($T_a$) and skin ($T_s'$) temperatures, and vertical temperature and humidity profiles. The real-time GEO retrievals employ National Centers for Environmental Prediction (NCEP) Global Forecast System (GFS; EMC, 2003) model forecasts accessed from the Man-computer Interactive Data Analysis System (McIDAS; Lazzara et al. 1999). Non-real-time GEO studies utilize either GFS or Modern-Era Retrospective Analysis for Research and Applications (MERRA; Reinecker *et al*., 2011) reanalyses. The impacts of using one reanalysis or the other are examined by analyzing the same satellite data using each of the two reanalyses during the $T_s$ retrieval. MERRA data have a spatial resolution of 0.5° latitude × 0.66° longitude over the globe. The surface skin temperature is available hourly, while the temperature and humidity profiles are provided every 6 hours. A total of 43 atmospheric layers are used. The version of GFS used here has a 1.25° horizontal resolution and up to 11 levels in the vertical, and provides data every 6 hours. No model values of $T_s'$ are available in the GFS version over land, so $T_s'$ is estimated from $T_a$ as a function of local time and season.





### 3 Single-channel skin temperature retrieval

The method for calculating $T_s$ from 11-µm $T_{toa}$ observations is an updated, higher-resolution version of that described by Scarino et al. (2013). Because some imagers (e.g., AVHRR-1, GOES-13) lack split-window capabilities, the single-channel method best allows historical consistency in application amongst many distinct sensors (Sun and Pinker, 2003; Jiménez-Muñoz and Sobrino, 2010; Heidinger, 2013). The analysis employs the cloud mask algorithm developed for the Clouds and the Earth's Radiant Energy System (CERES) to classify pixels as cloudy or clear on a chosen grid (Minnis et al. 2008b). The algorithm relies on comparisons of observations with estimates of the clear-sky $T_{toa}$ or reflectance at 0.65, 3.8, and 10.8 µm. Those estimates are made using the CERES 10' clear-sky albedo and land surface emissivity databases (Chen et al., 2004, 2010), along with the appropriate bidirectional and directional reflectance models, angularly dependent sea surface emissivity models, predicted skin temperature, and corrections for atmospheric absorption and emission (Minnis et al., 2011). The emissivity for water surfaces is estimated using a wind-speed-dependent model developed from theoretical calculations using the approach of Jin et al. (2006). A constant wind-speed of 5 knots is assumed for all pixels.

The observed or modeled radiance at the TOA can be represented as:

$$B(T_{TOA}) = \prod_{i=n}^{1} t_i \left[ B(T_o) \right] + (1 - t_1) B(T_1) + \sum_{i=n}^{2} (1 - t_i) B(T_i) \prod_{j=i}^{1} t_j , \tag{3}$$

where $T_o$ is the surface-leaving radiant energy equivalent brightness temperature, which comes from $T_s$ based on the following relationship using the narrowband surface emissivity:

$$T_s = B^{-1} \left\{ \frac{\left[ B(T_o) - (1 - \varepsilon_s) \times L_\downarrow \right]}{\varepsilon_s} \right\} , \tag{4}$$

where $L_\downarrow$ is the downwelling radiant energy at the surface:

$$L_\downarrow = (1 - t_n) B(T_n) + \sum_{i=1}^{n-1} (1 - t_i) B(T_i) \prod_{j=n}^{i+1} t_j , \tag{5}$$

The subscripts $i$ and $j$ denote an atmospheric layer, where 1 and $n$ refer to the layers at the TOA and just above the surface, respectively [e.g., $B(T_1) \equiv B(T_{toa})$]. The atmospheric layer temperature is $T_i$, and $B$ is evaluated at the central wavelength of the 11-µm band. $B^{-1}$ is the inverse Planck function. The layer transmissivity ($t_i$) derives from the correlated $k$-distribution technique. This technique is described in detail by Goody et al. (1989) and Kratz (1995), which depict the discrete version of the spectral-mean transmission $t_{\Delta\omega}(u,p,\theta)$ as:

$$t_{\Delta\omega}(u,p,\theta) \cong \sum_{i=1}^{n} w_i \exp \left[ -k_i(p,\theta) u \right] , \tag{6}$$

where $k_i(p,\theta)$ is an absorption coefficient as a function of pressure $p$ and temperature $\theta$ for a particular wavenumber $\omega$, $u$ is a pathlength, and $w_i$ is a weighting factor for which the summation over $n$ calculations must equal 1.

The surface temperatures and atmospheric profiles are linearly interpolated temporally to the satellite image time and spatially to the center of each 0.5°×0.5 AVHRR or 1.0°×1.0° GEO region. In the case of AVHRR retrievals, regions can have resolutions up to 1.5°×1.5° near the poles, but are nominally 0.5° × 0.5° everywhere else. The same $T_s$ retrieval methodology is used for all resolutions. The specific logic of the cloud mask algorithm can be found in Minnis et al. (2008a, 2010, 2016) and Trepte et al. (2010), which describe cloud tests for different scenarios (e.g., scenes over snow or desert, sun-glint-influenced ocean, scenes with smoke or thin cirrus). It is important to note that although the NWP skin temperature $T_s'$ is used as a seed value in the initial application of the cloud mask, decisions based solely on the difference between 11-µm observations and model values occur for only 2.3% (5.3%) of the pixels over land during the day (night). Therefore, the initial influence $T_s'$ is significantly diminished.



After the cloud mask is applied, the mean 0.65-µm reflectance and 3.8- and 10.8-µm $T_{toa}$ (i.e., $<T_{toa}>$) values are computed from the clear and cloudy pixels for each region. The data are then analyzed as $8 \times 12$-pixel tiles for AVHRR or $1.0° \times 1.0°$ regions for GEO. If at least 20% of the pixels within the tile or region are considered clear, the mean observed clear-sky temperature replaces the original NWP-based clear-sky temperature for the region and the cloud mask is repeated using the observed clear-

sky mean brightness temperature. The 20% criterion is used to minimize the influence of cloudy pixels on the final temperature value while still allowing sufficient sample size. If fewer than 20% of the pixels are clear, then the original clear-sky estimate $T_s'$ and cloud mask are retained and no value $T_s$ is retrieved.

For those tiles/regions satisfying the 20% criterion, a value of $T_s$ for each pixel is determined using a two-step process. First, the tile/regional mean value $T_s$ (i.e., $<T_s>$) is determined by solving Eq (3) from the inverse of Eq (4) (i.e, $T_o'$ solved from $T_s'$), and

then using the mean observed 11-µm clear-sky $<T_{toa}>$ to adjust $T_s'$ based on the difference between the $<T_{toa}>$ and the modeled $T_{toa}'$ for each tile/region. That is, a correction is applied to the model $T_s'$ and temperature/humidity profiles such that $T_{toa}'$ computed with Eqs (3) and (4) equals $<T_{toa}>$, thereby yielding $<T_s>$. For the AVHRR retrievals, $<T_{toa}>$ represents a tile value, whereas $T_{toa}'$ represents the larger regional value because the latter originates from the region-scale MERRA $T_s'$. Thus, all tiles having their center within a given MERRA grid box use the same model profiles and $T_s'$. For the GEO $T_s$ retrieval, both the

observed $<T_{toa}>$ and the modeled $T_{toa}'$ are represented on the $1.0° \times 1.0°$ region scale.

To save computational time, a value of $T_s$ is estimated for each pixel in the tile or region as

$$T_s = B^{-1}\left[R_T B\left(T_{TOA}\right)\right],$$

(7)

where $R_T$ is the ratio

$$R_T = \frac{B\left(\langle T_s\rangle\right)}{B\left(\langle T_{TOA}\rangle\right)},$$

(8)

and $T_{toa}$ is the observed clear-sky brightness temperature for the pixel. This approach yields $T_s$ pixel values that differ by $-0.04 \pm 0.20$ K from the $<T_s>$ computed using Eqs (3) and (4).

## 4. Sea surface temperature validation

Sea surface temperatures were retrieved as described above for the 2008 AVHRR datasets and are compared with the OI SST values. The AVHRR SST pixel data were first gridded to match the NOAA OI SST 0.25° resolution. Only those pixels classified

as clear, with 100% water fraction (based on a $1.0° \times 1.0°$ land mask) and 0% sea ice fraction outside of sun-glint conditions were used to compute the daily grid averages. Additionally, each pixel must be assigned a quality assurance flag of 1, indicating that there are no adjacent cloudy pixels or nearby thin cirrus (within two pixels).

Figure 3 maps the July 2008 SST means from AVHRR (Fig. 3a) and NOAA OI SST (Fig. 3b), and their differences (Fig. 3c), which qualitatively reveal very good agreement between the two products. The Fig. 3d scatter density plot reveals a more

quantitative analysis of the ~3 million daily, cell-to-cell comparisons. The bias and standard deviation of the difference (SDD) of the AVHRR SST relative to OI SST for July 2008 are -0.06 K and 0.62 K, respectively. A high associated coefficient of determination ($R^2 > 0.99$; not shown) indicates low variance, despite apparent outliers. Disagreements over open ocean, such as those in the tropical western Pacific and northern Pacific Ocean, can be attributed to cloud-clearing differences between the two products, or to the fact that the OI satellite SST is supplemented by in situ measurements from buoys and ships that are free of

cloud consideration. Nevertheless, despite localized coastal differences and cloud influences, the AVHRR SST is largely consistent with the NOAA OI SST product.



Sea surface temperatures from JAJO 2013 GOES-13 are compared to NOAA OI SSTs under the same gridding, filtering, and quality assurance criteria used for the AVHRR comparisons. Whereas the AVHRR SST retrievals always utilize atmospheric corrections based on MERRA reanalysis, the GEO SST retrievals utilize either GFS or MERRA for the atmospheric corrections. The near-real-time GEO retrievals currently rely on GFS forecasts, whereas the MERRA reanalysis is suitable for historical GEO

and AVHRR retrievals. Therefore, it is important to quantify the influence of the particular NWP reanalysis on satellite-based SST retrieval. Figure 4 compares the July 2013 GOES-13 SSTs retrieved using the GFS-based atmospheric corrections. The GOES SSTs are rather poor in both accuracy and precision relative to the reference – an absolute bias approaching -0.7 K with SDD = 1.02 K. These significant increases in bias and SDD can be attributed to the difference in NWP source, as is evident from Fig. 5. Figure 5 shows the same comparison as Fig. 4, except that MERRA profiles were used for the atmospheric corrections.

Similar then to the AVHRR retrievals, MERRA-derived GOES-13 SSTs exhibit a near-zero bias and an SDD of only 0.60 K relative to the NOAA OI SST reference.

The accuracy and precision of the GFS- and MERRA-derived GOES-13 SST values for the remaining seasonal months of 2013 are illustrated in Fig. 6 along with their AVHRR counterparts for all 12 months of 2008. Mean AVHRR SST is consistently 0.1-K, or less, colder than the NOAA OI SST reference throughout the year. The AVHRR SST monthly SDD is steady near 0.6 K.

The MERRA-based JAJO GOES-13 SST SDD is also steady near 0.6 K and the bias is consistently close to zero. The differences between the AVHRR and GOES biases are likely due to uncertainties in the infrared calibrations. The GFS-derived GOES-13 mean SST is consistently ~0.6 K colder than the NOAA reference, with an SDD in excess of 1.0 K for the JAJO seasonal months. This discrepancy with the MERRA-based results suggests that the GFS model profiles are drier than MERRA and/or have insufficient vertical resolution to properly account for the changes in water vapor that are used to compute the

atmospheric attenuation of the infrared radiation. It is unlikely that the GFS humidity is too low since it appears to have a wet bias (Yoo, 2012). An explanation for the differences in the model fields is beyond the scope of this paper. However, it is clear that the single-channel retrieval method is sensitive to the source of temperature and humidity profiles. Hereafter, the MERRA data are used for all analyses, unless indicated otherwise.

## 5. Land surface temperature viewing angle dependency correction

Satellite-observed LST depends on the viewing and illumination conditions because shading, vegetation conditions, soil type, and topography affect the radiance exiting the scene (Lagouarde et al., 1995; Minnis and Khaiyer 2000; Minnis et al. 2004). This thermal radiation anisotropy can result in the retrieved LST varying by 6 K or more for some areas (Rasmussen et al., 2010, 2011; Guillevic. et al. 2013). From experimental measurements, Sobrino and Cuenca (1999) and Cuenca and Sobrino (2004) found a VZA dependence of LST that depends on soil type. Pinheiro et al. (2006) developed a physical model to estimate the

variation of LST as a function of canopy coverage, solar zenith angle (SZA), VZA, and relative azimuth angle (RAA) for a savanna. Rasmussen et al. (2010, 2011) developed and applied a similar model to predict the LST that would be retrieved by Meteosat over Africa. Vinnikov et al. (2012) constructed a generalized model to correct for all angles, but, for general application, it requires many sets of matched measurements from different angle sets to construct the necessary kernels. Addressing the anisotropic effects, and thereby leading to more accurate interpretation of $T_s$, can not only improve climate

studies, but can also be of significant benefit to data assimilation and numerical weather prediction needs (Reichle et al., 2010; Guillevic et al., 2013; Draper et al., 2014).

Accounting for 3-D radiance anisotropy for a global retrieval methodology will require the development of regional and seasonal kernels for a universal model (e.g., Vinnikov et al. 2012) or developing canopy configurations globally for physical models (e.g.,



Rasmussen et al. 2010). In lieu of developing a comprehensive model that accounts for all three angles, a simple empirical model is developed in this section to account for the average dependence of the retrieved LST values on VZA. Although this initial step toward a universally applicable model provides correction for only one component of the anisotropy, it reduces the uncertainty of the retrieved LST values, as shown below.

**5.1. Nadir-Normalization Model**

The VZA correction model relies on simultaneous matched LST values from GOES-13 and Aqua-MODIS. Land surface temperature is retrieved from MODIS using the same single-channel methodology described in Section 2. Hourly JAJO 2013 GOES-13 and twice-daily Aqua-MODIS LST values were first averaged on a 1.0° × 1.0° grid encompassing 60°N to 60°S and 120°W to 30°W. For each MODIS overpass, the MODIS LST means of each gridbox were matched with their GOES

counterparts whenever the sampling times differed by less than 30 minutes. The GOES-East satellite was chosen to build the empirical model given the abundance of landmass in both the northern and southern hemispheres within the sensor field of view (FOV). The bulk of the higher VZAs, however, is concentrated in North America as the South American landmass dwindles south of 20°S. It is assumed that the VZA correction based on the GOES-East model will yield similar improvements when applied to LST retrievals from other satellites.

The MODIS LST retrievals limited to views having cosine(VZA), or $CVZA$, greater than D=0.95 serve as the nadir LST reference at all GOES-East viewing angles. That is, GOES-13-minus-MODIS mean LST biases are computed for intervals of VZA difference ($\Delta VZA$), expressed, however, in terms of cosine($\Delta VZA$), or $C\Delta VZA$, intervals. Here, because the MODIS data are from near-nadir views, $C\Delta VZA$ and $CVZA$ are essentially interchangeable. The empirical relationship is illustrated in Fig. 7 for daytime LST matches during the JAJO seasonal months. The $\Delta VZA$ interval step is 1°, increasing from a starting point of $\Delta VZA$

= 15°. In an effort to effect gradual, stable change in bias with respect to $C\Delta VZA$, a $C\Delta VZA$ allowance ($A$) was employed at each $C\Delta VZA$ interval ($I$) as follows:

$$A(I) = 0.5\left[\left(ID + \sqrt{(1-I^2)(1-D^2)}\right) - \left(ID - \sqrt{(1-I^2)(1-D^2)}\right)\right],$$
(9)

which derives from

$$A(\Delta VZA) = 0.5\left[\cos\left(\Delta VZA - \cos^{-1}(D)\right) - \cos\left(\Delta VZA + \cos^{-1}(D)\right)\right].$$
(10)

Thus, the mean bias (Fig. 7 solid black dots) is determined at each $C\Delta VZA$ interval $I$ for matched GOES-13 and MODIS LST pairs for which the satellite $C\Delta VZA$ is within the range $ID \pm A(I)$. The linear regression (Fig. 7 blue lines) of the mean GOES-minus-MODIS LST bias with respect to $C\Delta VZA$ constitutes the nadir-normalized VZA correction model,

$$\Delta T = a_1 C\Delta VZA + a_0,$$
(11)

where $a_1$ and $a_0$ are regression coefficients. The regression is tuned such that, for perfect nadir matching (i.e., $C\Delta VZA = 1$ or

$\Delta VZA = 0$), the $\Delta T$ adjustment is zero. This modification is made in order to prevent GOES-13 and Aqua-MODIS calibration differences (less than 0.35 K on average) from contributing false biases to the $C\Delta VZA$ dependency. That is, the bias difference between GOES and MODIS should only be a function of increasing VZA difference, and therefore must not be influenced by pre-existing calibration bias, which does not contribute to $C\Delta VZA$ dependency. The $\Delta T$ adjustment applied to satellite-derived skin temperature at $CVZA$, i.e., $T_{s,sat}(CVZA_{sat})$, yields nadir-normalized satellite-derived skin temperature, i.e., $T_{s,sat}(CVZA_{nadir})$, as

follows:

$$T_{s,sat}\left(CVZA_{nadir}\right) = T_{s,sat}\left(CVZA_{sat}\right) - \Delta T\left(CVZA_{sat}\right).$$
(12)



Note that although the model is designed for application with $CVZA$, it can be referred to as a VZA adjustment or correction for simplicity.

Further criteria beyond $C\Delta VZA$ stratification were required for including matched GOES-13 and MODIS pairs in the analysis. Both the GOES-13 and MODIS $T_s$ averages for each $1.0° \times 1.0°$ grid cell must be based on, at least, 125 pixels. For the mean

bias at a given $C\Delta VZA$ interval to qualify for use in the model, that interval regression must consist of at least 75 individual matched grid cells with SDD < 2.5 K. The sample limits of 125 and 75 were found to be an acceptable balance between sample-per-interval allowance and model stability. That is, these criteria are necessary to minimize uncertainties in the model while maximizing the $C\Delta VZA$ dynamic range.

A systematic relationship between increasing viewing angle and negative GOES- minus-MODIS daytime LST bias is evident for

April, July, and October (Figs. 7b-7d). During the day, as the GOES VZA increases ($CVZA$ decreases), the GOES-retrieved LST decreases relative to the nadir MODIS-retrieved LST. During the day in January (Fig. 7a) and at night for all months (Fig. 8), the viewing angle dependency of LST is virtually non-existent. Nighttime anisotropy effects, induced either by differential cooling (Minnis and Khaiyer, 2000) or varying emissivity contributions caused by different fractional amounts of vegetation in the sensor FOV (Vinnikov et al., 2012), were not detected in this empirical approach. The result in Fig. 8, however, is not

unexpected. Pinheiro et al. (2006) and Guillevic et al. (2013) found nighttime LST to be independent of viewing considerations, and conclude along with Minnis and Khaiyer (2000) that the primary cause of anisotropy is shadowing, with lesser contributions from evaporative cooling and surface air temperature gradients. The lack of a daytime VZA dependency during January (Fig. 7a) is surprising given that the average SZAs are larger in the Northern Hemisphere during January. It is evident, however, that the daytime VZA dependency is greatest when the sun is highest in North America. Whether this seasonal dependency is

representative of variations around the globe will require much additional analysis beyond the scope of this paper.

The anisotropy correction for the SatCORPS SST employs a wind and viewing angle-dependent sea surface emissivity model based on theoretical calculations using a 5-knot wind speed (Jin et al. 2006). The results in Fig. 9 demonstrate that the model is quite effective at minimizing the dependence of SST on VZA. The bias is essentially zero across the VZA range and is accompanied small SDD. Therefore, as with nighttime land cases, an additional nadir-normalization model appears to be

unnecessary for ocean scenes.

The dependence of the LST can also be expressed in terms of the variation of surface emissivity with VZA. This is illustrated in Fig. 10, which plots the ratios of the mean surface-leaving-radiances ($L$) from GOES-13 (GE) to those from the matched near-nadir Aqua-MODIS, along with linear regression fits of the data as a function of the $C\Delta VZA$. In this case, the surface emissivity would be characterized as

$$\varepsilon_s(C\Delta VZA) = \varepsilon_s\left(b_1 C\Delta VZA + b_0\right).$$
(13)

The slopes, $b_1$, and offsets, $b_0$, are shown in Fig. 10 for each month. As expected from Fig. 7, the slope is negligible for January (Fig. 10a) and is greatest for April (Fig. 10b). The apparent nonlinearity for $CDVZA < 0.6$ during April and July (Fig. 10c) may be an artifact of the reduced sampling at the higher VZAs (see Fig. 7).

## 5.2 Validation with independent MODIS LST, MYD11

The JAJO 2013 GOES-13 LST values are compared with the independent MYD11 product to determine if the VZA parameterization improves the consistency of the two products. The GOES-13 10.8-μm channel was first cross-calibrated as in Minnis et al. (2002) against its Aqua-MODIS counterpart, channel 31, to minimize any calibration differences. Spectral differences were taken into account as in Scarino et al. (2016), but are based on Infrared Atmospheric Sounding Interferometer spectral measurements. For each Aqua overpass, the MYD11 pixel LST values are averaged on the 1° x 1° GOES-East domain



and matched to within 15 minutes of the GOES-13 hourly scans, provided there are at least 150 valid MODIS and GOES-13 pixels per grid cell. To eliminate any differences due to surface emissivity discrepancies, the GOES-13 LST was retrieved using the MYD11 11.0-μm emissivity values. To effect the comparisons, the GOES-13 LST values were normalized to the MYD11 view, i.e., $CVZA_{myd}$, to yield $T_{s,sat}(CVZA_{myd})$ as follows:

$$T_{s,sat}\left(CVZA_{myd}\right) = T_{s,sat}\left(CVZA_{sat}\right) - \left[\Delta T\left(CVZA_{sat}\right) - \Delta T\left(CVZA_{myd}\right)\right].$$

(14)

The appropriate model in Fig. 7 is used to compute $DT$ in Eq (14) for each of the 4 months of matched data.

Figure 11 shows histograms of the differences between the GOES-13 and MYD11 LSTs without (Fig. 11a) and with (Fig. 11b) the daytime VZA corrections. Without correction, the GOES LSTs tend to be slightly greater than their MYD11 counterparts, especially during the daytime. The SDD is less than 1.9 K for all scenes, and the day and night GOES biases are 0.42 K and 0.11 K, respectively, resulting in a combined (both day and night) 0.28-K bias. After applying Eq (14) using the daytime corrections from the appropriate months, the daytime SDD and bias drop to 1.76 K and 0.08 K, respectively. The nocturnal bias drops to almost -0.2 K, while its SDD increases slightly. Applying the daytime corrections at night does not improve the comparisons, although the mean combined bias of -0.04 K is somewhat closer to zero. If the daytime correction is used during day only, the combined bias and SDD are 0.09 K and 1.49 K, respectively. It is clear that normalizing the VZAs of the two retrievals yields better agreement during the day, but it appears that no VZA correction is needed at night for the MYD11 data, as in accordance with the Fig. 8 results.

Similar results (not shown) were found for the GOES-13 LST values retrieved using GFS instead of MERRA. Unlike the SST comparisons (Figs. 4 and 5), the GFS-derived GOES LST bias and SDD values are comparable to those based on the MERRA profiles. Without applying the VZA corrections, the nocturnal and daytime biases for GOES/GFS retrievals relative to MYD11 are 0.14 ± 1.12 K and 0.54 ± 2.07 K, respectively, which are not significantly worse than the corresponding MERRA values. After applying the VZA adjustment, the night and day biases are -0.15 ± 1.18 K and 0.20 ± 1.93 K, respectively. Although the GFS results over land, compared to ocean, are much closer to those from MERRA, the MERRA-based results are slightly more accurate, relative to MYD11, than their GFS counterparts.

### 5.3 GOES-East/West LST comparison

To further test the efficacy of the VZA corrections, differences between the hourly GE and GOES-West (GW) LST retrievals from July 2013 were computed before and after applying the daytime July VZA adjustment. Prior to differencing, the 15-minute discrepancy in the image retrieval at the 3-hourly synoptic times (00, 03, …, 21 UTC) was mitigated by adjusting the GE LST, which is based on images beginning 15 minutes before the UTC hour, to that UTC hour when the GW image scan began. This approach accounts for the specific GE and GW scanline time discrepancies. The GE data were linearly interpolated to the GW time using the nearest surrounding synoptic hours. When those surrounding hours crossed the sunrise terminator, no correction was applied because of the day-night discontinuity in LST that occurs shortly after sunrise. Data taken near the terminator (solar zenith angle between 80° and 100°) were not used. The image times at the non-synoptic hours are nearly identical, so no temporal normalization was required. To minimize calibration differences, the average nocturnal LST difference, 0.08 K, between GOES-13 and 15 within 0.5° longitude of 105°W, which is bisector of the two views, was computed and added to all GOES-15 (GW) values.

Figure 12 plots the VZAs for GW (Fig. 12a), GE (Fig. 12b), and the GE − GW VZA differences (Fig. 12c). Although the differences are generally less than ±30°, the largest VZAs are up to 70° or more, so $C\Delta VZA$ can be as much as 0.35. Figure 7c would suggest large LST differences for pairs matched at the higher VZAs in this domain. All the retrieved values of normalized LST for both satellites were adjusted to nadir using the equation in Fig. 7c to account for the VZA dependence.



The mean regional differences, i.e., $DT_s$ = LST(GE) – LST(GW), are shown in Fig. 13 for the matched July 2013 data. During daytime, $DT_s$ for the unadjusted values (Fig. 13a) is mostly positive east of 105°W and negative to the west. Notable exceptions include the positive values in the west corresponding the highest mountain ranges in Colorado, Utah, Mexico, Washington, Wyoming, Idaho, and New Mexico. After adjusting to nadir (Fig. 13b), the same patterns remain, but the $DEW$ values are closer

to zero except for those in the high mountain areas, which are enhanced with the adjustment. Also, the corrected differences for some of the regions at extreme VZAs in the far northeast remain relatively large, perhaps because the viewing dependence increases with VZA with a greater slope for VZA > 60° as suggested by Fig. 10c. At night, the unadjusted differences (Fig. 13c) are relatively small, $|DT_s| < 2$, in most regions. The positive differences are no longer evident over the high mountains. Applying the daytime VZA correction further reduces $|DT_s|$ to values less than 1.0 K in nearly all cases (Fig. 10d).

Table 1 summarizes the GE – GW results. Over the eastern and western halves of the domain, $|DT_s|$ drops by 0.97 K and 0.55 K, respectively, during the day with the application of the VZA adjustment. The mean regional differences are much smaller than before correction, especially for the western region where the difference is near zero. Similarly at night, the corresponding regional differences decrease by comparable amounts and are much closer to zero than without the corrections. Furthermore, the mean absolute biases for both day and night, which are determined by the east – west sample-weighted region differences (not

shown), are much closer after correction – reduced by a factor of two or more. In contrast to the findings in the previous sections, there appears to be a dependence of $T_s$ on VZA over land at night, at least, for the VZAs seen here, which are mostly greater than 40°. The reasons for the discrepancy at night are not immediately evident and warrant additional investigation in future studies. Overall, the mean bias for the entire domain after correction over all non-terminator hours is 0.59 K.

Although it significantly reduces the GE – GW differences, the VZA correction does not eliminate all of the disagreement

between the two satellite retrievals. This is especially evident over the mountains. Also, although sign difference between the means over the eastern and western domains essentially disappears for both day and night with the correction, the remaining east-west difference suggests other factors aside from VZA affect the observed temperatures. It is likely that the solar azimuthal dependence seen in earlier studies (e.g., Minnis et al. 2004, Vinnikov et al. 2012) is not balanced out for the configurations seen here. The azimuthal dependence includes effects from both the relative solar azimuth angle and the azimuthal orientation of the

terrain and vegetation. Moreover, the heating/cooling rates probably differ between the eastern and western domains because of humidity and altitude differences. Downwelling longwave radiation might play a greater role in the diurnal cycle of $T_s$ in the eastern domain, perhaps diminishing the solar-induced anisotropy. Although the azimuthal dependencies are outside the purview of this paper, it is instructive to further explore how the differences change over the course of the day and how much the VZA correction diminishes the differences in more detail.

To that end, the differences were averaged for each UTC and are plotted in Fig. 14 as lines connecting the means at each hour. Over the western domain (red line), the uncorrected $DT_s$ (Fig. 14a) gradually approaches zero at 09 UTC from ~-1 K after 03 UTC, when the sun has set over the entire domain. At 12 UTC, it rises rapidly to a peak of 2.5 K near 16 UTC and drops precipitously after 17 UTC to -3 K at 22 UTC before increasing to 03 UTC. In the east (blue line), $DT_s$ drops slowly toward zero after 01 UTC, but only reaches 0.4 K at 06 UTC before increasing again. It only increases significantly after 12 UTC,

maximizing at 3.5 K (17 UTC) before decreasing to 1.3 K at 21 UTC, when it levels off. The relative behavior is the same for the corrected values (Fig. 13b), but the two curves are nearly identical between 03 and 17 UTC, being much closer to zero overall than without the VZA correction. After 17 UTC, the curves diverge with the western data changing more rapidly than their eastern counterparts suggesting different cooling rates. The bias for the entire domain (black line) shows definitively that the afternoon points are mainly responsible the overall positive bias in Table 1.



Even with different cooling rates, it is expected that $DT_s$ would approach zero after correction for VZA effects as the surface air and skin temperature equilibrate. Instead of going to zero after 03 UTC, $DT_s$ drops to roughly-0.3 K for the entire domain by 06 UTC and then rises to +0.5 K at 09 UTC, remaining flat until 12 UTC. This odd behavior is likely an artifact of the sun-satellite configuration, which causes a change in the infrared channel calibrations at satellite midnight and for 3-4 hours afterward. Yu et al. (2013) found that the GOES-11 and GOES-12 10.7-μm (channel 4) brightness temperatures were biased by -0.5 K relative to their daytime calibrations for 3-4 hours after satellite midnight, even after an operational correction for the midnight effect had been applied. A smaller bias was evident for a couple of hours prior to midnight. This residual bias could explain the unexpected variation in $DT_s$ seen between 03 and 12 UTC, if GOES-13 and 15 suffer from a similar UTC bias. Assuming then that the calibration biases are -0.15 K and -0.30 K two hours before and for four hours after midnight, respectively, for GE and -0.25 K and -0.5 K for GW, then $DT_s$ would almost follow the black curve in Fig. 14b exactly (assuming that $DT_s$ = 0 in a perfectly calibrated system). By 06 UTC, $DT_s$ would reach -0.30 K because only GE is influenced by the midnight effect. By 07 UTC, the smaller GW pre-midnight bias would partially offset the GE bias causing $DT_s$ to rise until 09 UTC, when only GW is affected. After 12 UTC, the daylight in the eastern half of the domain would overwhelm any remaining bias.

The results here only represent one domain during one month. The *CDVZA* functions in Fig. 7, the midnight calibrations, and the viewing and illumination angles vary with time of year. It is clear that much more comprehensive study would be needed to fully assess the VZA component of the angular dependence of the retrieved $T_s$ values.

**5.4 Validation with ground stations**

The ground sites used for further validation consist of the ARM SGP and seven SURFRAD locations: Bondville, IL (BON), Desert Rock, NV (DRA), Fort Peck, MT (FPK), Goodwin Creek, MS (GWN), The Pennsylvania State University (PSU), Sioux Falls, SD (SXF), and Table Mountain, CO (TBL). To obtain estimates of the LST bias and SDD relative to ground site measurements, all 3x3-pixel arrays centered on each site, having confidently clear LST values, were selected from the GOES-13, GOES-15, and NOAA-18 AVHRR retrievals. Averages and standard deviations were computed for each array. Any array with a standard deviation greater than the 99[th] percentile of standard deviations for each site was eliminated to minimize the inclusion of any residual cloudiness in the arrays. After screening, the LST of the central pixel in each array was used for comparison with the SURFRAD station measurements, while the array mean LSTs are compared with the SGP data. It was determined that, in general, the central pixel yielded better comparisons than the array means owing to terrain heterogeneity. For relatively homogenous regions such as the SGP and SXF, however, either value could be used.

Figure 15 shows the scatterplots of LST retrieved from the ARM SGP IRT and from matched GOES and AVHRR data. The IRT is a down-looking narrow-field-of-view instrument, so it is considered to have a nadir view for this comparison. The points (Fig. 15a) tend to parallel the line of agreement, but are mostly above it. The IRT values are 1.10-K greater than their satellite counterparts. The SDD is nearly 2.0 K. If the daytime VZA corrections are applied to all of the data, the points are scattered about the line of agreement and the average difference is 0.02 K with SDD = 1.78 K.  If comparing the central pixel rather than the 3x3 array average, the average difference is then -0.03 K with SDD = 1.68 K (not shown). With either scenario, the agreement improves for both daytime and nighttime points suggesting that the VZA dependency discussed in the previous section is valid for night also. This use of a daytime-based model for all hours may seem unattested, but given the conflicting results regarding the existence of nighttime VZA dependency, only the daytime model is available for assessing improvement at night. Overall, the results support its application to nighttime data.

This improvement for both halves of the diurnal cycle is easier to see in Fig. 16, which plots histograms of the differences, SatCORPS – IRT, before (Fig. 16a) and after (Fig. 16b) VZA correction. The daytime bias approaches zero, moving from -1.46





K to -0.30 K, while the nocturnal bias increases from -0.78 K to 0.30 K. The histograms narrow as SDD improves for both time periods. If only the GOES data are considered, the corrected data yield 0.03 ± 2.09 K and 0.28 ± 1.05 K for day and night, respectively. This suggests that AVHRR retrievals have slightly larger uncertainties during the day than the GOES retrievals. The same data were analyzed using the GFS atmospheric profiles and yielded smaller biases, -1.16, -0.63, and -0.86 K, for day,

night, and all times, respectively, for no VZA correction (not shown). With the correction, the day and night biases of 0.07 K and 0.55 K, respectively, which combine to yield an overestimate of 0.34 ± 1.55 K. This bias is slightly larger than the MERRA-based retrievals, but the SDD is reduced by 13%. Thus, the overall accuracy is similar for the two vertical profile sources for this location.

The SURFRAD down-looking pyrometers, used for estimating $T_s$ with Eq (2), are hemispherical sensors receiving radiation from

all directions. Thus, the equivalent VZA for comparison is 53°, which corresponds to the diffusivity factor of 1.66. Equation (14) is used to effect the VZA corrections of the satellite data, except that cos(53°) replaces $CVZA_{myd}$ in all of the appropriate terms. Because the viewing perspective of the SURFRAD sites from GE ranges from ~43°–63°, the VZA difference is typically < 10°, and as such most of the corrections will be relatively small. However, AVHRR views a given site over a wider range of VZAs, resulting in larger corrections for some overpasses.

Figure 17 shows the matched satellite and SURFRAD LSTs at Desert Rock, NV without (Fig. 17a) and with (Fig. 17b) the VZA adjustments applied. The change after applying the adjustments is negligible, as expected. During spring and winter, SatCORPS underestimates $T_s$ from the surface, while it tends to match well during the summer and autumn months. Overall, the bias is close to -1.0 K for this site, which is located in a valley surrounded by low mountains that rise more 300 m from the valley floor. The GOES and AVHRR pixels could include some portion of the surrounding terrain given that the mountains are within 2.5 – 2.7

km from the site. The mountains could induce a small difference between satellite and surface-site temperatures. The LST differences over Sioux Falls, SD are plotted in Fig. 18, with two different colors depicting day (red) and night (blue) points. Without the VZA correction (Fig. 18a), the bias is slightly negative, but becomes slightly positive after correcting for the VZA (Fig. 18b). In both cases, the SDD is 1.86 K. The increase in $T_s$ after correction brings the hotter daytime points in agreement, but overcorrects the colder points. The nighttime data are fairly well aligned in both cases. For this reference, there is minimal

elevation change within 10 km of the site.

Table 2 gives the mean bias and SDD for all satellites together (bottom) and for the GOES retrievals alone (top) for each SURFRAD site using MERRA as the SatCORPS input. The averages of the surface emissivities used at each site are also given. Bias and SDD results are shaded to show where the VZA correction yielded improved (green) or degraded (red) results by an absolute change of more 0.05 K or greater. Smaller differences are considered negligible. Brighter shades indicate an absolute

adjustment of 0.10 K or greater. The absolute biases for the combined day-night data are all less 1.0 K, except for TBL. Removing the AVHRR data from the comparisons (top) worsens the combined biases for SXF. However, most of the biases are smaller for the GOES-only results than for the combined satellites. For VZA-corrected GOES-only results, the ranges in SDDs are 1.21-2.10 K, 1.23-2.15 K, and 1.06-1.66 K for the combined, daytime, and nighttime sets, respectively. Adding the AVHRR data yields the corresponding ranges in SDD: 1.30-2.09 K, 1.28-2.36K, and 1.07-1.78 K. In general, the uncorrected SDDs are

larger than their corrected counterparts.

The biases in the results can be due to many factors including errors in the assumed surface emissivities, the atmospheric profiles, and the surface observations themselves. The representativeness of the site for the much larger area is also potentially a large source of bias. This issue, sometimes called the up-scaling problem (Li et al., 2014; Guillevic et al., 2012), is a concern for any ground-based satellite LST validation effort, but no attempt is made here to up-scale the ground station point observations to

fully characterize the relatively large pixel area of the satellite product. The potential impact of the large scale is important to





mention, however. For example, Guillevic et al. (2012) found that, in areas of mixed trees and open grass or farmland, tree canopy temperatures rise less during the day and drop less at night than the surrounding bare soil (open areas). Because the surface radiometers are typically located in open

areas, they could read systematically warmer or colder than the average upwelling radiation for the entire domain, which encompasses canopies and surface-exposed areas. This may, in part, explain the day-night switch in the sign of the bias over GWN and PSU – two sites with considerable mixtures of trees and open fields. Heidinger et al. (2013) found a similar day/night bias change for those two sites. Topography can also be responsible, in part, for the biases. PSU and DRA are both located in valleys around which the elevation changes by 300-m or more within the area represented by a single pixel. The large negative bias at TBL may be due, in part, to elevation change around the site. These and other error sources should be explored in detail in

future analyses.

The VZA adjustment generally improves the nocturnal biases and about half of the daytime biases. To quantify the improvement, the results from all of the SURFRAD sites were combined to produce the histograms in Fig. 19. Compared to the differences for the uncorrected data (Fig. 19a), the corrected SatCORPS temperatures (Fig. 19b) represent a slight improvement, day and night, in both bias and SDD, but produce a 0.04-K increase in absolute bias when all data are combined. Overall, the VZA-corrected

temperatures generate biases and SDDs of -0.51 K and 2.13 K for daytime, respectively, and -0.01 and 1.77 K at night. The 3% drop in SDD for the combined data represents an almost negligible improvement in the precision. As noted earlier, however, the GOES VZAs relative to these sites are near the diffusivity angle, and the change was expected to be small. Furthermore, for the AVHRR data, a full range of VZAs was used, so that, on average, the correction should also be small. The only location where the VZA corrections were significant was the ARM SGP site, where the data were corrected to a nadir view. That adjustment

was quite significant, taking the bias much closer to zero.

The GOES analyses were also performed using the GFS profiles as input. The mean differences and SDDs between all surface measurements, including the SGP site, and SatCORPS retrievals using both MERRA and GFS input are summarized in Table 3. Overall, the GFS soundings yield slightly larger SDDs and a greater positive nocturnal bias, which produces a smaller combined bias than found using the MERRA data. Although the LST retrievals are similar for both GFS and MERRA, they differ

significantly for SSTs (Fig. 6). Thus, use of the MERRA profiles for retrieving $T_s$ with a single IR channel is preferable. The combined day and night MERRA-sourced biases and SDDs are comparable to those of Heidinger et al. (2013) despite different assumptions and input, and are smaller than those from the current operational GOES product (Sun et al., 2012). Additionally, with the exception of daytime BON SDD, the individual site day and night MERRA-sourced accuracy and precision values, as well as GFS-sourced values with the exception of SXF SDD (not shown), are within the GOES-R specifications of 2.5 K and 2.3

K, respectively (Yu et al., 2010).

Heidinger et al. (2013) also reported very small changes in LST as a function of VZA and concluded that they are not a major concern. The VZA corrections developed here improve the absolute bias and SDD in nearly all cases (Table 3). Although the corrections can increase the bias at night, they reduce the SDD and absolute bias in the combined results. The VZA correction, on average, reduces the absolute bias and SDD by 0.2 K and 4%, respectively, for the 8 surface sites. These small improvements,

together with the better satellite-to-satellite normalization in Fig. 13, demonstrate that adjustment of LST for VZA dependencies will result in a more accurate and uniform product. The VZA dependency is probably not a concern for VZA < 45°, as suggested by Wan and Li (1997). However, as greater VZAs are used, particularly for GEOs, the VZA correction should be considered. The effects of the illumination angles on the retrievals, however, should also be taken into account as they clearly have the greatest impact during the day, as indicated by Fig. 14. That aspect of the retrieval problem remains for future study.





### 6. Summary and Conclusions

Accurate assessment of global climate and improvement of climate models, as well as numerical weather forecasts, rely on consistent land and ocean $T_s$ measurements, among others. Atmospheric flux calculations depend on the robustness of such surface variables, and NWP analyses are driven by reliable and frequent state variable updates over large spatial domains.

Despite key downsides, satellite data are ideal sources of $T_s$ given their model-ready retrieval schedule and broad continuous areal coverage. Thermal-infrared-derived $T_s$ relies on accurate cloud clearing, atmospheric adjustment, and viewing angle dependency correction. Therefore, validation of satellite $T_s$ relative to known standards is of critical importance.

The SatCORPS provides a $T_s$ product retrieved from GEO and AVHRR sources using the same single-channel algorithm. The benefit of the single-channel approach is that this method is more universally applicable to historic and future satellite

instruments compared to the split-window technique. Having GEO and AVHRR $T_s$ values derived from the same algorithm reduces relative uncertainty and, hence, are better able supplement one another. Validation of SST retrieved from both satellites demonstrates consistent accuracy and precision results of less than 0.1 K and 0.6 K relative to NOAA OI SST, respectively, for atmospheric corrections based on MERRA profiles. If GFS temperature and humidity profiles are used to account for atmospheric attenuation, however, the accuracy and precision values for the GEO SST exceed 0.6 K and 1.0 K, respectively. The

larger negative bias and precision relative to the MERRA-based results suggests that the GFS atmosphere is drier than MERRA over the oceans, on average. This result is surprising in that satellite (Tian et al., 2013) and radiosonde (Kennedy et al., 2011) comparisons indicate that MERRA is too dry at altitudes below 500 hPa.

Daytime LST retrievals can be significantly influenced by satellite viewing geometry. As such, a seasonally dependent empirical model was developed using nadir single-channel MODIS and GOES-13 LST retrievals to account for this angular dependency.

The model, which is formulated either as a temperature difference or emissivity adjustment, can be used to normalize a satellite LST to any viewing zenith angle. A nighttime VZA dependency was not observed, further supporting the idea that sun/shade discrepancy is the dominant driver of LST anisotropy. A January daytime VZA dependency was also not found, perhaps indicating the chief importance of emissivity differences resulting from terrain and canopy-surface configurations that change with VZA.

Land surface temperatures retrieved from July 2013 matched GOES-East and GOES-West data over North America showed distinct VZA-dependent differences. Normalization of the daytime LSTs to the nadir view using the July daytime correction model reduced the absolute bias by a factor of two. The remaining daytime differences are due to solar illumination effects that are not considered here. Despite the absence of any VZA dependence in the matched nighttime GOES – MODIS data, the GE – GW average nocturnal absolute LST difference is ~0.9 K. Applying the daytime VZA correction reduces the mean absolute bias

to ~0.2 K. The conflicting results from the two different satellite analyses may be the result of diminished sampling at high VZAs in the matched MODIS-GOES dataset, as most of the GOES-East/West pairs have VZA > 45°. This discrepancy in nocturnal VZA dependence between the two results should be examined further along with the midnight calibration effect, which causes a temporary bias in GEO LST for part of the night period.

The SatCORPS retrievals from GOES-13 were compared to the Collection-5 Aqua-MODIS LST product, a well-validated

dataset. Normalization of the daytime GOES LSTs to the MODIS VZAs reduced the bias and SDD by roughly 0.3 K and 0.1 K, respectively, bringing the GOES data to within 0.1 ± 1.8-K of the daytime MODIS product. Applying the daytime VZA normalization at night slightly worsened the comparison with MODIS, but when combined with the daytime data, it reduced the differences to nearly 0.0 ± 1.5 K. Use of the GFS profiles in place of their MERRA counterparts slightly degraded the precision to 1.6 K. Comparisons with LSTs from eight disparate ground stations provide further evidence of the validity of the SatCORPS

retrieval approach and the application of the VZA corrections, both for day and night. The VZA corrections increase the



accuracy by almost 50% and the precision by less than 10%, representing a net benefit. On average, MERRA-based atmospheric corrections seem to perform slightly better than GFS-based attenuation for LST retrievals compared to surface and other satellite LSTs. This finding, however, should not restrict use of GFS for LST retrievals, as the differences are rather small and not consistently better/worse in all scenarios. For SST validation, the MERRA atmosphere is clearly preferred.

This study has examined data from only one small part of the Earth over a limited range of angles for the VZA model development. It is not clear that a VZA-correction method developed for scenes over North America is applicable to other locations and the appropriate seasons. There remains some question about the behavior of the dependence at VZA > 45°, particularly at night. Many land areas are viewed at high VZAs by GEO imagers and, therefore, a more comprehensive characterization of the VZA dependence is warranted as satellite-to-satellite differences will produce climatological artifacts if

the VZA dependence is not mitigated.

Further investigation is warranted for the SURFRAD validation approach, especially for the GWN, TBL, and PSU locations, and particularly in terms of the up-scaling problem. Disparity between pixel- and SURFRAD-observed surface conditions and topography-induced model sounding deficiencies are likely contributors to the surface-satellite differences. Beyond the outlier cases, however, the SatCORPS GEO and AVHRR $T_s$ exhibit high accuracy and precision, with VZA-normalization affording

reductions of 0.2 K and 0.1 K in daytime absolute LST bias and SDD, respectively. By incorporating these near-global NRT retrievals, the data assimilation and climate research communities will hopefully benefit from improved forecasts and better understanding of the global climate model.

**Acknowledgments**

This research was supported by the NASA Modeling, Analysis, and Prediction Program and the NOAA CDR Program.
Computing was supported by the NASA High End Computing Program. The authors would like to thank Sarah Bedka and Doug Spangenberg for their generous assistance with MODIS skin temperature processing.

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





| Longitude | Day | | Night | | All | |
|---|---|---|---|---|---|---|
| | Before (K) | After (K) | Before (K) | After (K) | Before (K) | After (K) |
| < 105°W | 2.38 | 1.41 | 1.18 | 0.29 | 2.07 | 1.10 |
| > 105°W | -0.51 | 0.04 | -0.55 | 0.00 | -0.53 | 0.02 |
| All | 0.98 | 0.74 | 0.43 | 0.17 | 0.85 | 0.59 |

**Table 1:** July 2013 matched GOES-East minus GOES-West mean clear-sky surface skin temperature difference for regions east and west of 105°W. The sample-weighted average bias is shown in the bottom row.



| Sites | | | BON | | DRA | | FPK | | GWN | | PSU | | SXF | | TBL | |
|---|---|---|---|---|---|---|---|---|---|---|---|---|---|---|---|---|
| | | | 40.1°N, 88.4°W | | 36.6°N, 116.1°W | | 48.3°N, 105.1°W | | 34.3°N, 89.9°W | | 40.7°N, 77.9°W | | 43.7°N, 96.6°W | | 40.1°N, 105.2°W | |
| Mean surface emissivity | | | E11 | ELW | E11 | ELW | E11 | ELW | E11 | ELW | E11 | ELW | E11 | ELW | E11 | ELW |
| | | | 0.985 | 0.981 | 0.982 | 0.954 | 0.992 | 0.986 | 0.983 | 0.989 | 0.984 | 0.990 | 0.991 | 0.981 | 0.089 | 0.994 |
| Data used and sampling | | | Temperature Differences (K) | | | | | | | | | | | | | |
| | | | Orig | Corr | Orig | Corr | Orig | Corr | Orig | Corr | Orig | Corr | Orig | Corr | Orig | Corr |
| GOES Only | All | Bias | 0.67 | 0.51 | -0.97 | -0.88 | -0.33 | 0.14 | 0.83 | 0.48 | 0.81 | 0.52 | -0.04 | 0.21 | -1.81 | -1.71 |
| | | SDD | 1.61 | 1.61 | 1.37 | 1.37 | 1.25 | 1.21 | 2.05 | 2.10 | 2.02 | 2.06 | 1.94 | 1.93 | 1.90 | 1.91 |
| | Day | Bias | 1.11 | 0.93 | -1.36 | -1.19 | -0.25 | 0.24 | -1.00 | -1.37 | -0.53 | -0.84 | 0.10 | 0.39 | -2.35 | -2.25 |
| | | SDD | 2.07 | 2.09 | 1.62 | 1.60 | 1.27 | 1.23 | 1.43 | 1.52 | 1.70 | 1.72 | 2.16 | 2.15 | 2.04 | 2.05 |
| | Night | Bias | 0.35 | 0.21 | -0.74 | -0.70 | -0.53 | -0.10 | 2.26 | 1.93 | 1.84 | 1.57 | -0.25 | -0.06 | -1.37 | -1.27 |
| | | SDD | 1.07 | 1.06 | 1.15 | 1.18 | 1.18 | 1.13 | 1.08 | 1.11 | 1.60 | 1.63 | 1.56 | 1.54 | 1.65 | 1.66 |
| GOES + AVHRR | All | Bias | 0.99 | 0.80 | -0.94 | -0.96 | -0.07 | 0.26 | 0.83 | 0.51 | 0.88 | 0.56 | -0.07 | 0.06 | -1.67 | -1.65 |
| | | SDD | 1.84 | 1.84 | 1.57 | 1.52 | 1.45 | 1.30 | 2.00 | 2.03 | 2.06 | 2.09 | 1.86 | 1.86 | 2.08 | 2.04 |
| | Day | Bias | 1.42 | 1.25 | -1.00 | -0.95 | -0.13 | 0.29 | -0.90 | -1.22 | -0.65 | -0.97 | -0.02 | 0.19 | -2.36 | -2.28 |
| | | SDD | 2.35 | 2.36 | 1.97 | 1.79 | 1.40 | 1.28 | 1.46 | 1.55 | 1.76 | 1.79 | 2.11 | 2.11 | 2.23 | 2.18 |
| | Night | Bias | 0.69 | 0.48 | -0.91 | -0.97 | 0.04 | 0.21 | 2.23 | 1.91 | 1.90 | 1.56 | -0.14 | -0.09 | -1.15 | -1.17 |
| | | SDD | 1.29 | 1.27 | 1.20 | 1.28 | 1.54 | 1.35 | 1.06 | 1.07 | 1.54 | 1.60 | 1.49 | 1.47 | 1.78 | 1.78 |

**Table 2:** Combined (bottom half only) mean bias and SDD values in Kelvins for each of the seven SURFRAD locations during day, night, and combined day and night conditions, before (Orig) and after (Corr) VZA correction. Top half: GOES-East and GOES-West. Bottom half: GOES-East, GOES-West, and AVHRR. Shades of green (red) indicate bias or SDD improvement (degradation) after applying the daytime VZA correction, for absolute changes greater than or equal to 0.05 K. Bright shades of green or red indicate an absolute change of at least 0.10 K. Coordinates and mean 11-µm (E11) and broadband longwave (ELW) emissivities are also listed for each site.



| MERRA | GOES | | GOES Corrected | | All | | All Corrected | |
|---|---|---|---|---|---|---|---|---|
| | Bias | SDD | Bias | SDD | Bias | SDD | Bias | SDD |
| Day | -0.73 | 2.13 | -0.41 | 2.07 | -0.71 | 2.24 | -0.47 | 2.15 |
| Night | -0.14 | 1.69 | 0.09 | 1.59 | -0.07 | 1.74 | 0.05 | 1.67 |
| Combined | -0.42 | 1.94 | -0.16 | 1.85 | -0.37 | 2.01 | -0.17 | 1.93 |
| GFS | Bias | SDD | Bias | SDD | Bias | SDD | Bias | SDD |
| Day | -0.73 | 2.17 | -0.43 | 2.08 | - | - | - | - |
| Night | 0.10 | 1.74 | 0.37 | 1.63 | - | - | - | - |
| Combined | -0.30 | 2.00 | -0.02 | 1.90 | - | - | - | - |

**Table 3:** Mean bias and SDD values based on results from the ARM and seven SURFRAD sites before and after VZA correction using only GOES data, and using both GOES and AVHRR results (All). SatCORPS retrievals based on MERRA (top) and GFS (bottom) input.





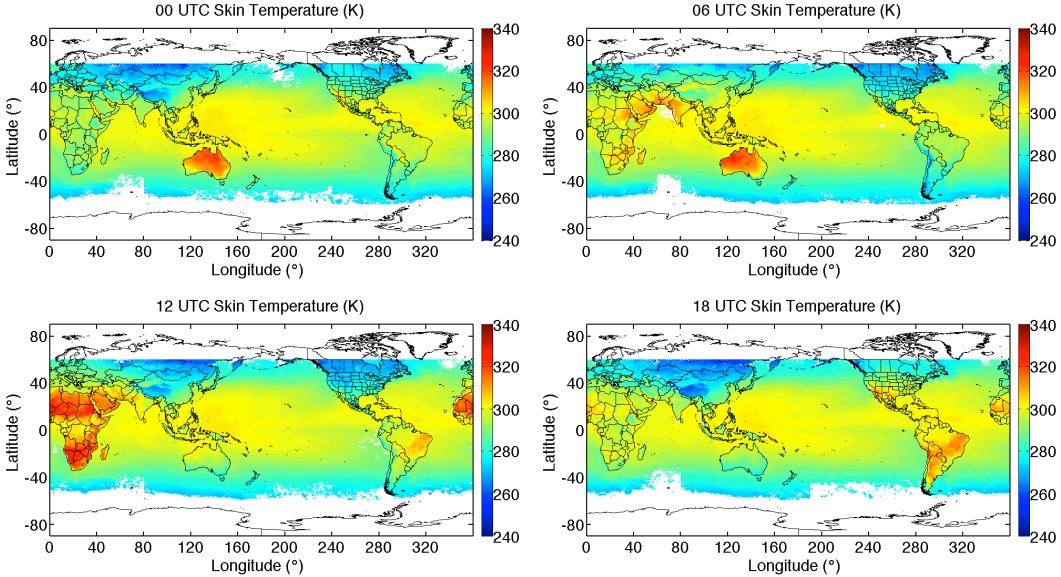

**Figure 1:** Mean merged, clear-sky surface skin temperature values from GOES-East, GOES-West, Meteosat-9, MTSAT-2, and INSAT-3D, October 2015.



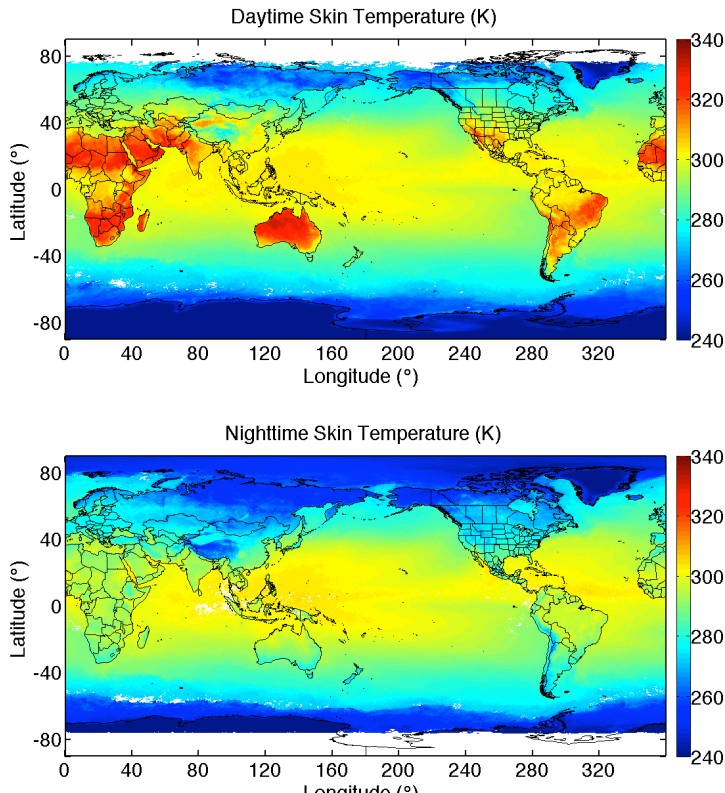

**Figure 2:** Average surface skin temperature from NOAA-18 AVHRR, October 2008.





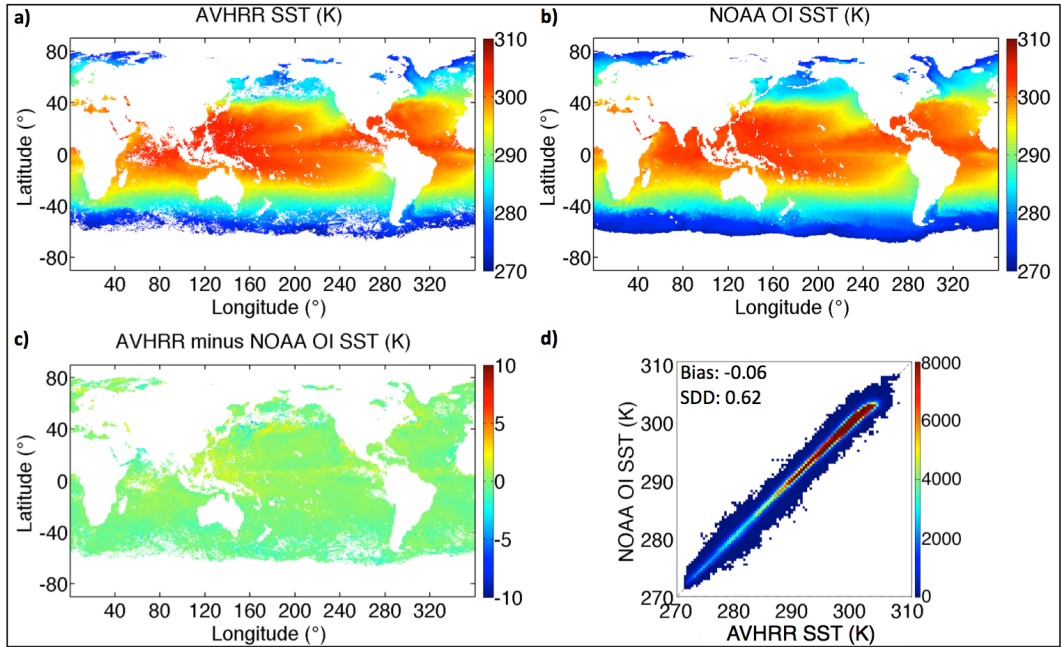

**Figure 3:** July 2008 a) AVHRR SST, b) NOAA OI SST, c) SST difference, and d) scatter density analysis of ~3 million daily matched grid

5    cells.



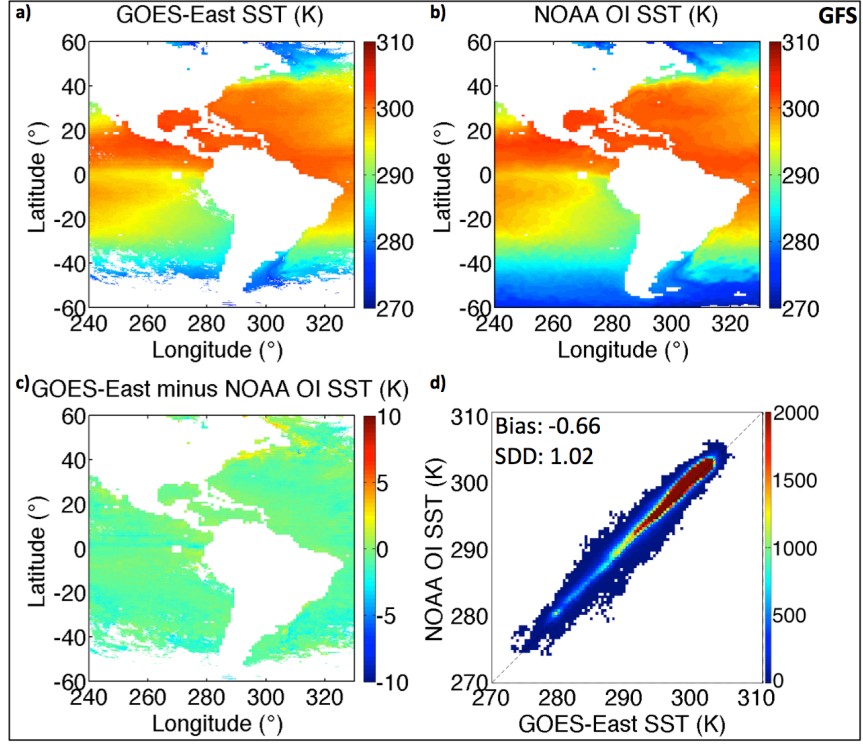

**Figure 4:** July 2013 a) GOES-13 SST derived, in part, from GFS-based atmospheric corrections, b) NOAA OI SST, c) SST difference, and d) scatter density analysis of ~1 million daily matched grid cells.



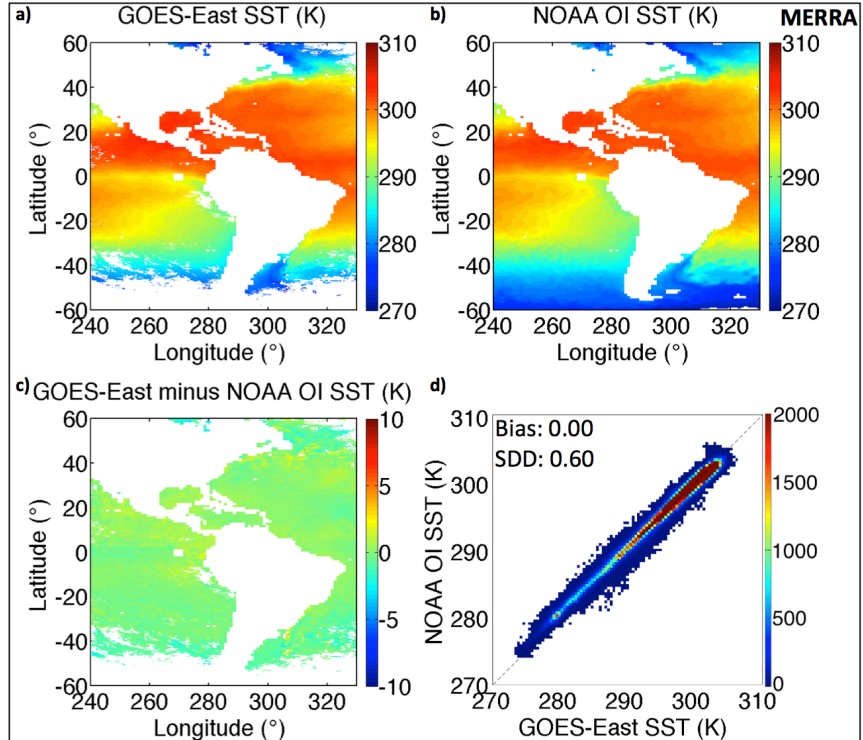

**Figure 5:** July 2013 a) GOES-13 SST derived, in part, from MERRA-based atmospheric corrections, b) NOAA OI SST, c) SST difference, and d) scatter density analysis of ~1 million daily matched grid cells.





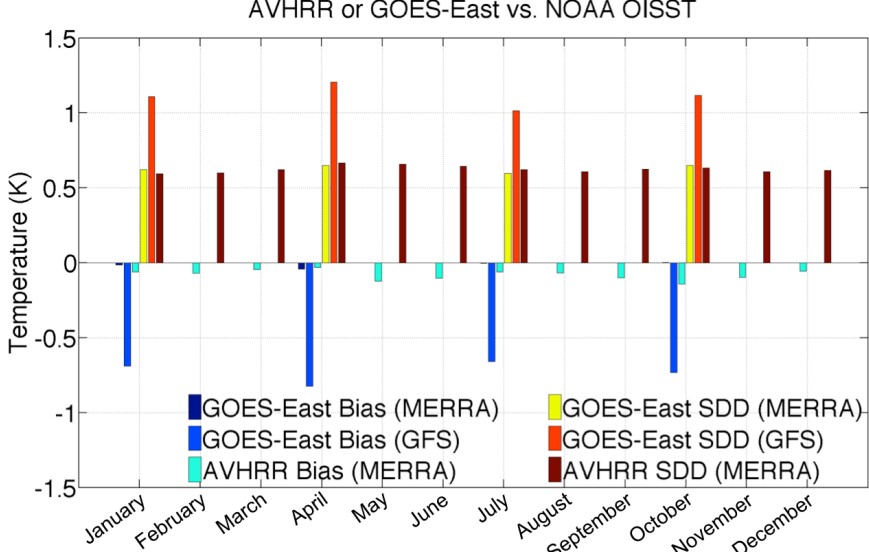

**Figure 6:** AVHRR (2008) and GOES-13 (2013) SST accuracy and precision relative to NOAA OI SST. For the GEO retrievals, the atmospheric correction is based on either GFS or MERRA reanalysis. Atmospheric corrections for AVHRR retrievals are strictly based on

5   MERRA.





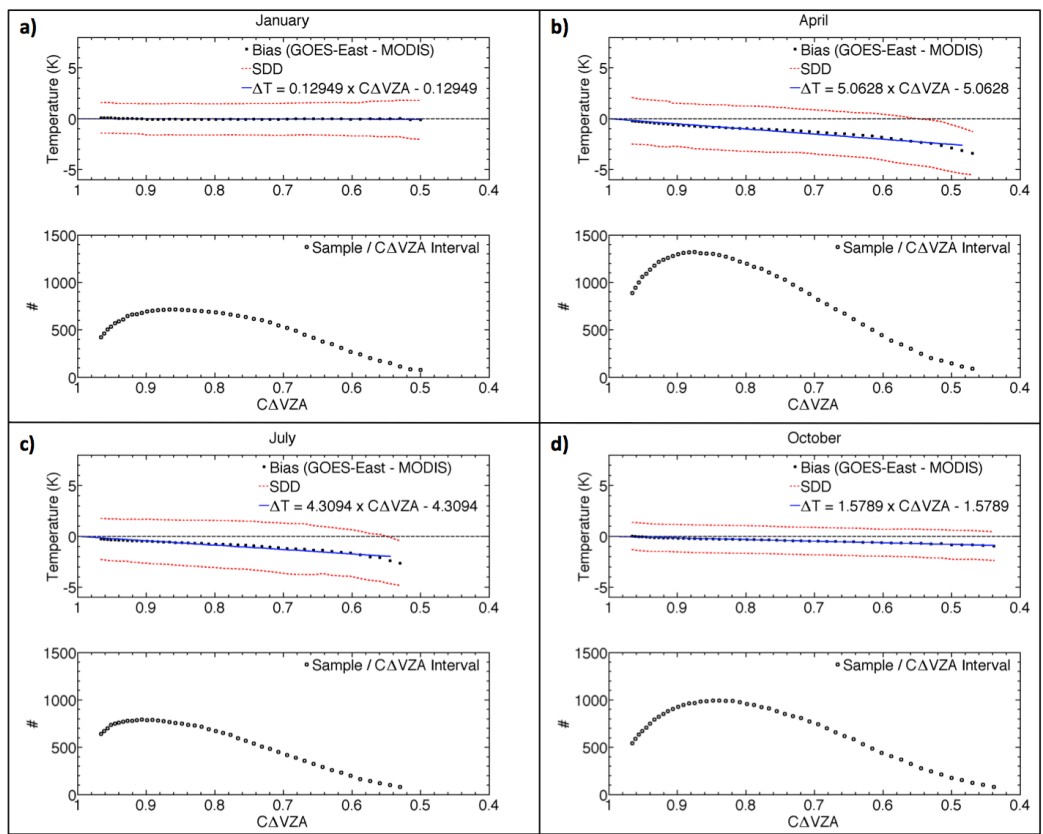

**Figure 7:** Daytime GOES-13-minus-MODIS LST bias (solid black dots) as a function of the cosine of the GOES-13-minus-MODIS VZA difference, for which MODIS retrievals are restricted to the nadir view, during a) January, b) April, c) July, and d) October 2013. The standard deviation of the difference (SDD) is indicated by the dotted red line. Black circles indicate the number of coincident GOES-13 and MODIS measurements at each *CΔVZA* interval.



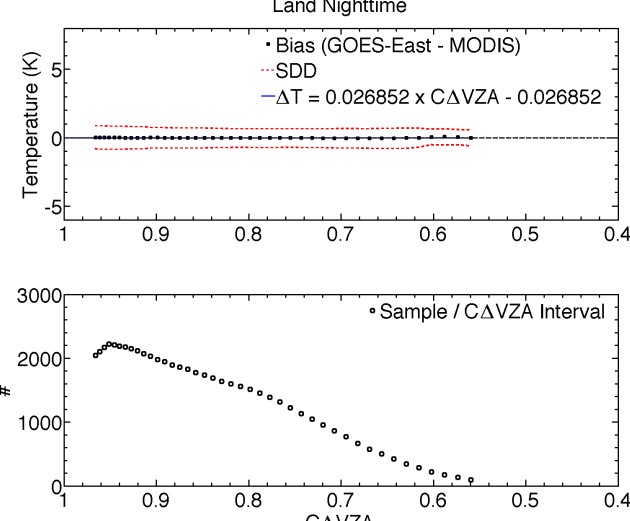

**Figure 8:** Same as Fig. 7, except for combined JAJO nighttime LST.





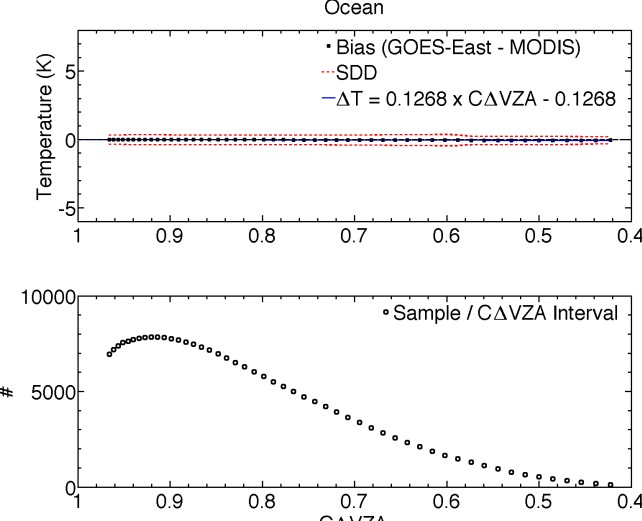

**Figure 9:** Same as Fig. 7, except for combined JAJO daytime and nighttime SST.





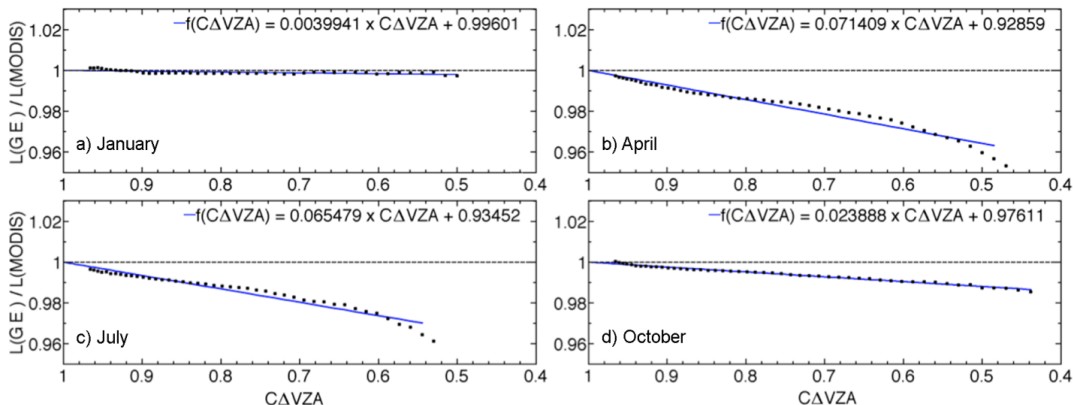

**Figure 10:** Same as Fig. 7, except for ratio of GOES-13 to Aqua-MODIS surface-leaving radiance. Sampling is the same as in Fig. 7.




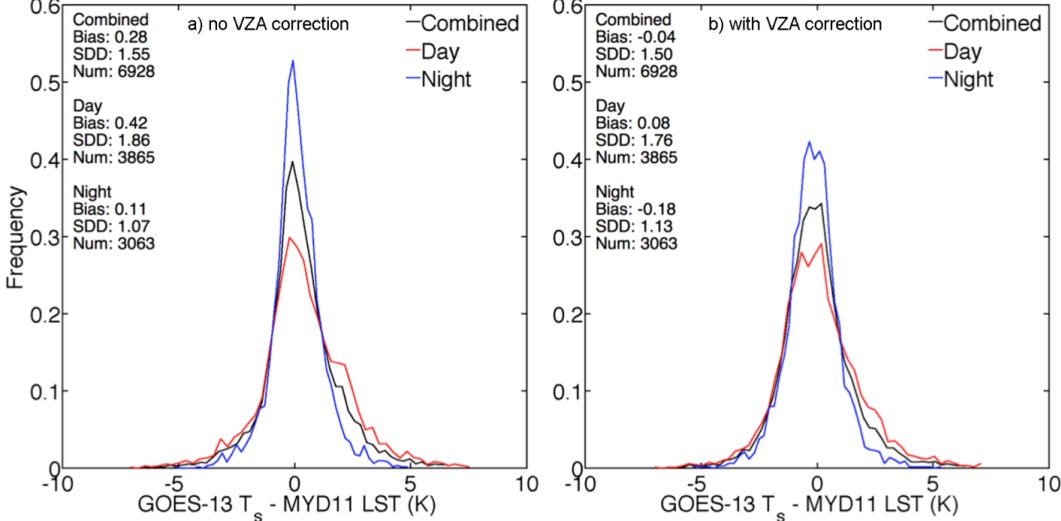

**Figure 11:** Probability distributions of LST differences from GOES-13 and the MYD11 Aqua-MODIS product for day, night, and all times (combined) (a) without and (b) with daytime viewing angle adjustments applied.

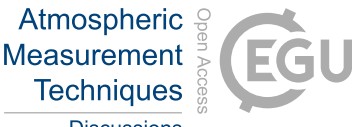



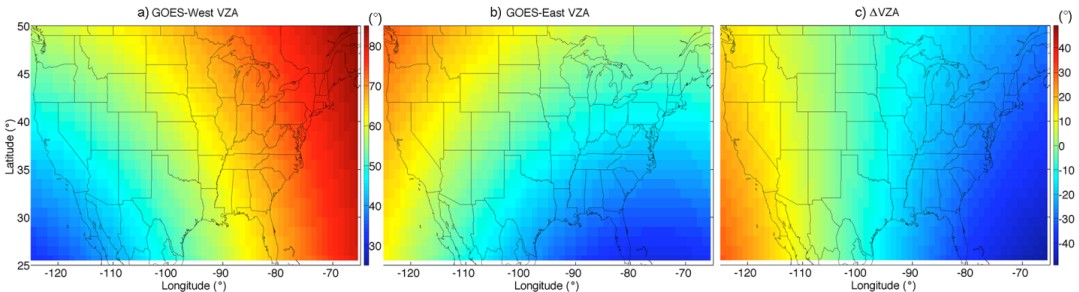

**Figure 12:** Viewing zenith angles for (a) GOES-West and (b) GOES-East, and (c) their differences over the matching domain.





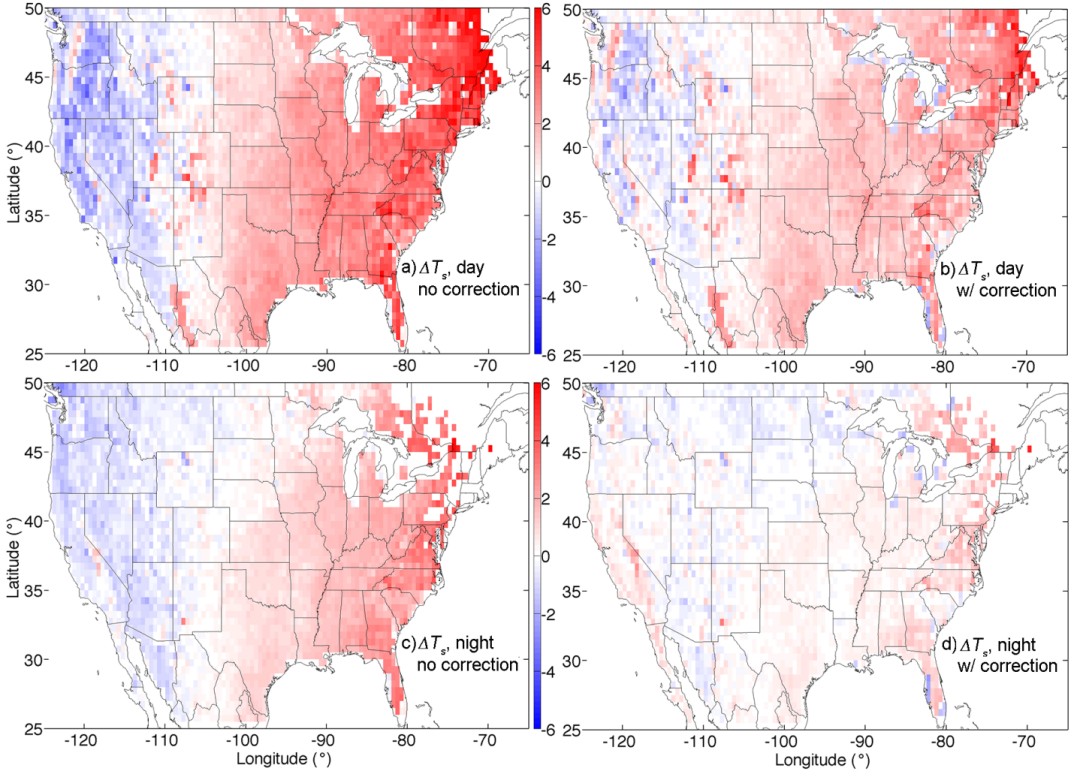

**Figure 13:** Mean regional GOES-East – GOES-West LST differences for July 2013. (a) Day and (c) night, no VZA adjustment. (b) Day and (d) night, with VZA adjustment.



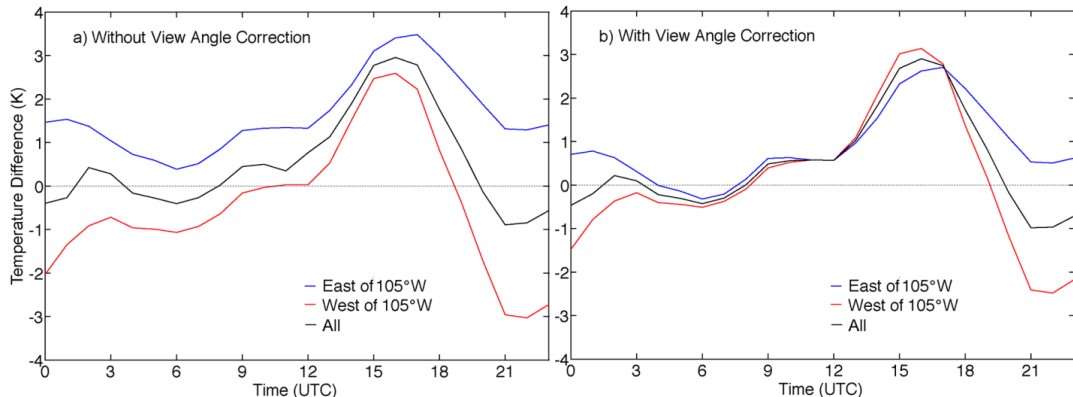

**Figure 14:** Mean hourly, regional GOES-East – GOES-West LST differences for July 2013.



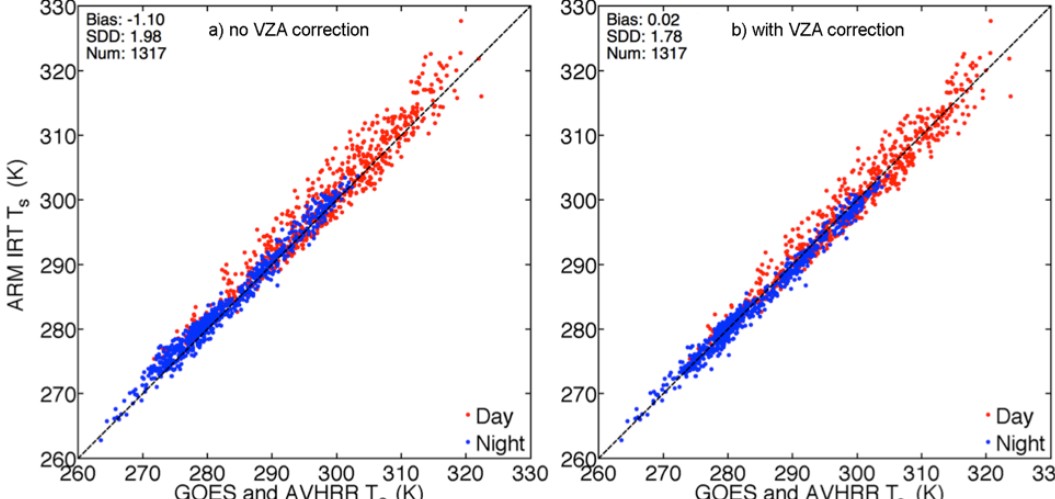

**Figure 15:** Scatterplots of clear-sky surface skin temperatures from JAJO 2013 GOES-13 and 15 imagery, and from 2008 NOAA-18 AVHRR data, matched with ARM SGP IRT temperatures (a) without and (b) with VZA corrections.



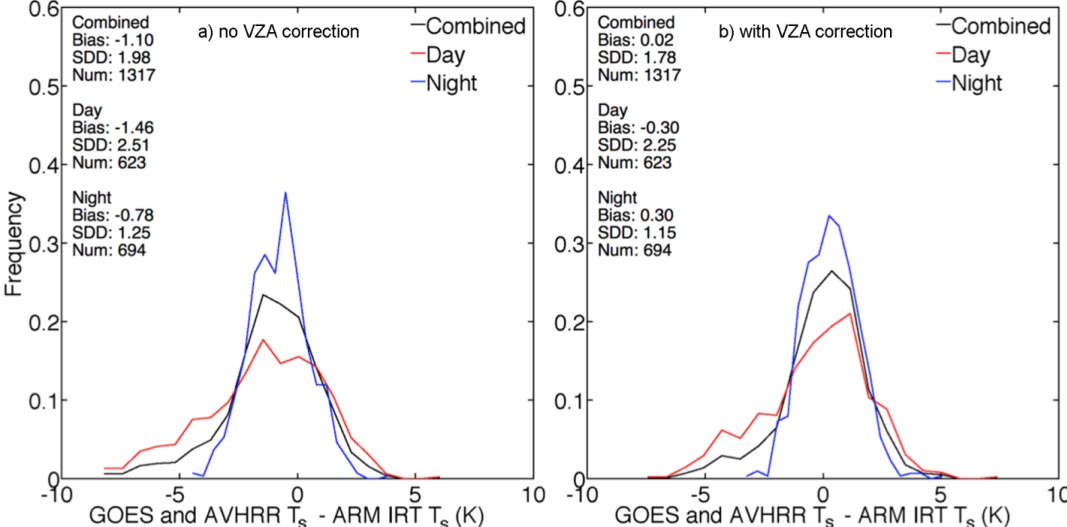

**Figure 16:** Histograms of LST differences between satellite (GOES-13, GOES-15, and AVHRR) and ARM IRT (a) without and (b) with VZA corrections.





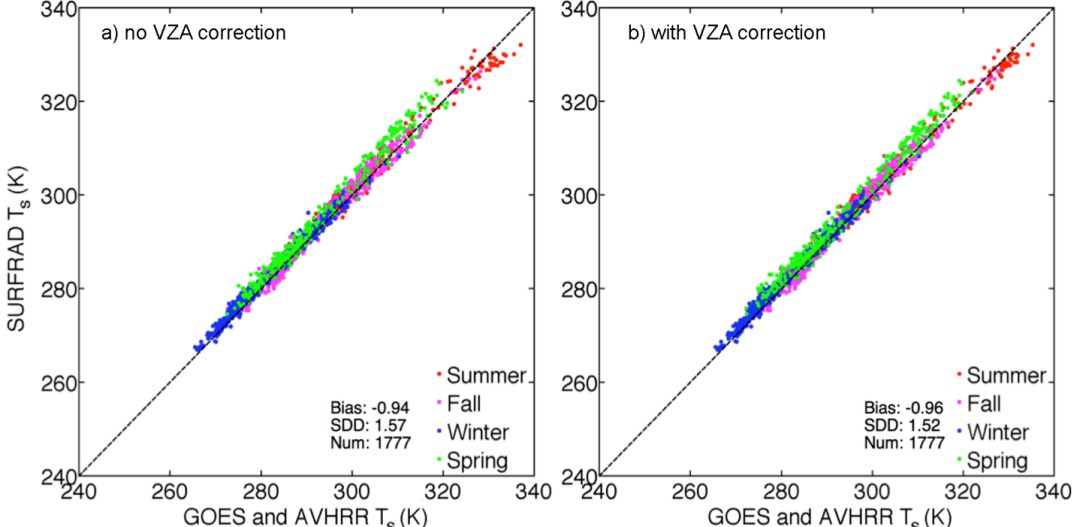

**Figure 17:** Scatterplots of LST from matched satellite (GOES-13, GOES-15, and AVHRR) and SURFRAD data at Desert Rock, NV, (a) without and (b) with VZA corrections.





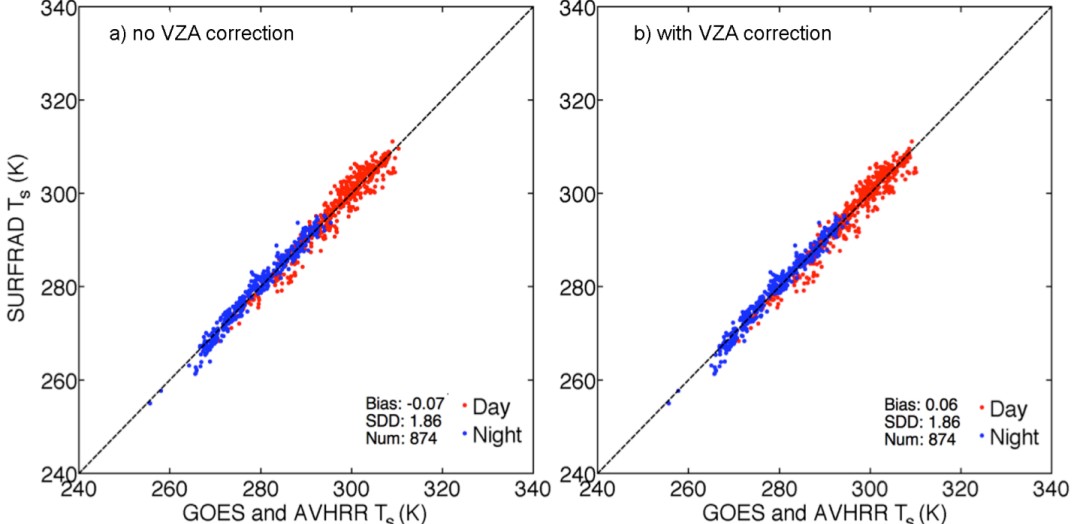

**Figure 18:** Same as Fig. 17, except for data over Sioux Falls, SD.





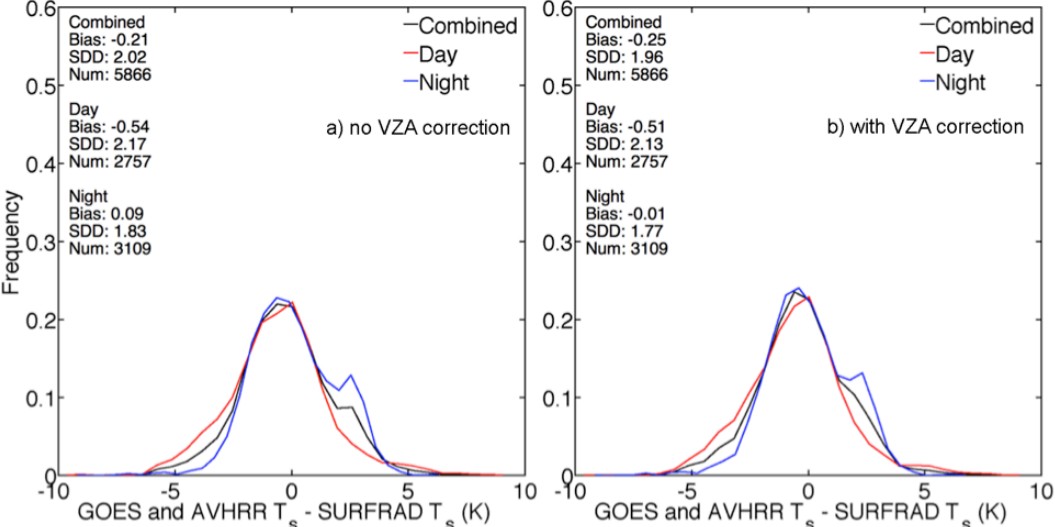

**Figure 19:** Differences between GOES-13, GOES-15, and AVHRR day, night, and combined LST and SURFRAD LST (a) without and (b) with VZA corrections.