# Peer review of "Global clear-sky surface skin temperature from multiple satellites using a single-channel algorithm with viewing zenith angle correction"

_Atmospheric Measurement Techniques, 2016_

## Referee Comment (RC1) · Anonymous Referee #1 · 24 May 2016

The article describes a methodology to derive Land Surface Temperature (LST) using observations from a single channel in the thermal infrared, which can be applicable to various sensors. On top of this, the authors propose a simple model to correct angular effects on satellite LST, allowing its correction to nadir view. The latter together with a large set of validation results form the main novelty of the current manuscript. The study is of interest for AMT and the manuscript is overall well presented. The results and in particular the LST angular correction require, however, further discussion. Following my comments below, this manuscript should be subject to major revisions before being considered for publication.

1) My main concern refers to the angular correction suggested in this article and its

interpretation. The characterization of LST angular effects is based on LST estimates from GOES and MODIS-Aqua, using the same algorithm and collocated in space and time. The authors then derive global regressions shown in Fig.7, where the correction depends only on the deviation of the view zenith angle to nadir. Results in Fig. 7 suggest that the coefficients of (daytime) regressions have a strong seasonality. I find those monthly fits to be strongly dependent on the seasonality observed over northern America, where most land pixels with high view angles are. The authors seem to somehow acknowledge this fact (lines 19-20 page 9), but discard its main implication, i.e., that those angular corrections may be applied everywhere. The validation over land (where angular effects are more relevant) is performed for ground sites over North America. I have strong doubts whether similar corrections would yield similar results in other sites (e.g., mid-latitudes in South America).

2) The angular effects on LST depend on surface types (vegetation density, orography), illumination angles, which in turn strongly influence temperature contrasts among surface elements within each pixel. These may translate into a simple model based on viewing angle deviations from a reference view, changing with local seasons. As such, "The lack of a daytime VZA dependency in January" (lines, 17-18, page 9) is in fact not surprising, as temperature differences among sunlit/shadowed surfaces are usually much lower in winter than in summer. If no other variables are taken into account, I do not see how a simple angular correction as that proposed in this article may be derived for the full GOES disk, including pixels with the same viewing angle in the Northern and Southern Hemispheres.

3) Following the point above, the results presented in fig. 11 (GOES versus MYD11), which I suppose cover the whole GOES disk, suggest that no night-time correction is needed; this is in line with fig.8. However, comparisons made for ARM and SURFRAD sites show improvements after angular corrections performed for night-time LST, which often surpass those observed for daytime. Were such night-time corrections made using the "daytime adjustment"? How can we physically explain such outcome?

4) AVHRR results are never shown separately from GOES-13 results. The article must include a description of the "AVHRR-only" validation and on the applicability of the angular correction to AVHRR observations, particularly to those obtained with larger (e.g., > 40°) view zenith angles. The same period of data should be used for validation of GOES and AVHRR LST to exclude the influence of inter-annual variability when comparing those results. The stability of angular corrections over different years should also be taken into account.

5) Minor/editorial comments:

a. Why using MODIS-Aqua as opposed to characterizing the angular effects with MODIS-Terra or with both Terra and Aqua?

b. Fig. 3c, 4c, 5c: the scale does not seem adequate to SST comparisons. Please reduce the scale max/min to at least +/-5K.

c. In section 5.4, it is indicated that the matchups for some ground stations consider the nearest pixel only (heterogeneous areas), while for others a 3x3 array is used. For the sake of simplicity, I strongly suggest the same criterion to be used for all.

d. Fig. 14 clearly shows the influence of illumination angles. Given the large difference between UTC and local time, can you indicate night-time periods (e.g., east and west of 105°, respectively)? It would also be interesting to see these differences for other periods of the year in a multi-panel fig.

e. Table 2: Correct TBL sfc emissivity for 11 micro-m.

f. Missing references: Ghent et al. (2010); Duan et al. (2014); Wang et al. (2014); Coll et al. (2009); Williamson et al. (2013); Wan et al. (2002); Li et al. (2014); Yu et al. (2010); Sobrino and Raissouni, 2000

g. Please discriminate the two references to Yu et al (2012) in your reference list and in the text; Sobrino et Romaguera (2004) not referred in the text.

---

## Referee Comment (RC2) · Anonymous Referee #2 · 6 Jun 2016

Major comments:

1. The major concern in this article is the inclusion of the ground station analysis. The article should not be considered fit for publication with these results included. I would suggest removing these results (reasons provided below) and with careful attention to other comments below and from other reviewers it would be fit for publication after major revision.

Although a few authors have used SURFRAD and other FLUXNET ground stations for LST 'validation', detailed analysis of these sites by Wang et al. 2009 and Guillevic et al. 2012 show that they are in fact unsuitable for validation of sensors at the kilometer or more scale resolution. Fluxes from these sites are measured with pyrgeometers on

10m towers giving them an effective spatial footprint of 30-45 meters, which compared to GOES at 4km (scaled to 8km) is essentially a point measurement. Considering that surface skin temperatures can vary a few degrees over distances of a few meters, this is simply not a valid comparison. e.g. Wang et al. 2009 found that large surface heterogeneity at these sites (e.g. Bondville surroundings go from fully veg to bare soil within a few meters of the tower, and Desert Rock has clumps of much warmer dark maffic rocks a few hundred meters from the tower) accounts for 60–70% of the error when making comparisons between ground-based measurements and LST retrievals from ASTER (90m). Guillevic et al. 2012 concluded that only by using an upscaling model to account for these heterogeneities was it possible to make any kind of validation assessment with kilometer-scale data. Wang et al. 2009 concluded SURFRAD sites should only be used with nighttime data from high-res sensors at 100-m scale.

Therefore claiming that there is an VZA improvement at these sites is really quite meaningless considering the overwhelming number of uncertainties based on site variability, scale difference, and emissivity estimation. There is no discussion on how emissivity was estimated at the SURFRAD sites? It is critical to get in situ emissivity measurements from the PI's themselves at these sites, given their fine-scale variability, and a simple assumed land cover classification will not suffice.

2. To first order the average view angle dependent correction is a move in the right direction but without dependence on elevation and surface type information it is hardly a complete correction, and may result in compensating errors dependent on pixel location.

3. Pinheiro et al (2006) and Guillevic et al (2013) have already shown that the nighttime LST is independent of the viewing considerations, so the relevancy of bias correction at night with this methodology is questionable. You are likely compensating for GOES-MODIS time differences or spatial aggregation of MODIS LST.

4. There is no discussion or mention of the possible effects of aggregating MODIS

1-km to the effective GOES pixel resolution (assumed 4 km here?). Temperature does not scale up in a linear fashion (emissivity does), so you are introducing an additional uncertainty in your VZA corrections from the scaling.

5. Eq. 13. Without temperature/emissivity separation you can not simply imply an emissivity dependence from the surface leaving radiance fraction, which makes this section and all accompanying figures invalid assumptions.

Please double-check references (e.g. Wang et al. 2014) is not included.

---

## Author Comment (AC1) · 28 Jun 2016

The comment was uploaded in the form of a supplement:
http://www.atmos-meas-tech-discuss.net/amt-2016-79/amt-2016-79-AC1-supplement.pdf

---

## Author Comment (AC2) · 28 Jun 2016

**Author Response**

**Thank you for taking the time to review and offer feedback for our manuscript. Our responses are offered below. You will also find details on where to find changes in the manuscript (if applicable) that are in reply to your comments. These changes are highlighted in yellow or green. Please note that other highlighting colors correspond to changes prompted by other reviewers.**

1. The major concern in this article is the inclusion of the ground station analysis. The article should not be considered fit for publication with these results included. I would suggest removing these results (reasons provided below) and with careful attention to other comments below and from other reviewers it would be fit for publication after major revision.

Although a few authors have used SURFRAD and other FLUXNET ground stations for LST 'validation', detailed analysis of these sites by Wang et al. 2009 and Guillevic et al. 2012 show that they are in fact unsuitable for validation of sensors at the kilometer or more scale resolution. Fluxes from these sites are measured with pyrgeometers on 10m towers giving them an effective spatial footprint of 30-45 meters, which compared to GOES at 4km (scaled to 8km) is essentially a point measurement. Considering that surface skin temperatures can vary a few degrees over distances of a few meters, this is simply not a valid comparison. e.g. Wang et al. 2009 found that large surface heterogeneity at these sites (e.g. Bondville surroundings go from fully veg to bare soil within a few meters of the tower, and Desert Rock has clumps of much warmer dark maffic rocks a few hundred meters from the tower) accounts for 60–70% of the error when making comparisons between ground-based measurements and LST retrievals from ASTER (90m). Guillevic et al. 2012 concluded that only by using an upscaling model to account for these heterogeneities was it possible to make any kind of validation assessment with kilometer-scale data. Wang et al. 2009 concluded SURFRAD sites should only be used with nighttime data from high-res sensors at 100-m scale.

Therefore claiming that there is an VZA improvement at these sites is really quite meaningless considering the overwhelming number of uncertainties based on site variability, scale difference, and emissivity estimation. There is no discussion on how emissivity was estimated at the SURFRAD sites? It is critical to get in situ emissivity measurements from the PI's themselves at these sites, given their fine-scale variability, and a simple assumed land cover classification will not suffice.

**It is critical that we quantify how our results relate to comparisons performed in the past. There is a precedent in the literature, a standard that has been established, of conducting validations in this manner, as you have acknowledged. Although Wang and Liang 2009 and Guillevic et al. 2012 led studies focused on the scaling problem, new literature has been published since those that provide non-scaled validation results relative to the very same**

ground sources, e.g., Heidinger et al. 2013 and Guillevic et al. 2014. Even though Guillevic et al. 2014 recommend scaling, they do not say SURFARD sites cannot be used, and in fact offer both scaled and non-scaled results. Then there is Heidinger et al., who recognize "[a] major source of uncertainty in comparing surface estimates of LST with satellite-derived estimates is the difference in spatial scales...[t]he satellite sensor provides an area-effective LST that can differ significantly from the LST from a single surface site in regions of surface heterogeneity." They continue to acknowledge that "[t]he comparisons of the satellite and SURFRAD results will therefore underestimate the impact of this potential error source. Users of these results... should be aware of this issue." Like Heidinger et al., we derive a single-channel LST from GOES and AVHRR and compare to, effectively, point-source-like ground measurements, and like them, we acknowledge the potential negative impact of up-scaling, and the need for further analysis (page 14, lines 12-30). The magnitude of that potential negative impact is questionable, however, as Wang and Liang 2009 concluded that biases were similar when comparing 3x3 (270 m) and 11x11 (1000 m) ASTER LST pixel averages in their assessment of LST variability at six SURFRAD ground sites.

We are very comparable to Heidinger et al., to whom it is necessary that we assess ourselves given the obvious similarities in our design. Furthermore, we can attest to that high comparability with confidence only for the reason that in neither of our studies is there an attempt to up-scale. Additionally, as previously mentioned, Guillevic et al. 2012 and Guillevic et al. 2014 highlight both non-scaled and up-scaled results, and with confidence we can assess our findings with their non-scaled comparisons, which we surpass. In fact, our mean difference and standard deviation results are arguably more comparable to the Guillevic et al. 2012 Table 5 *Satellite vs. scaled-up LST* results than the *Satellite vs. non-scaled LST* for both day and night, although this is not stated in our text. Finally, the fact that the standard deviations and mean differences we provide are reduced in response to application of VZA dependency corrections indicates that we are drawing closer to the reference measurement, which should serve as logical verification that uncertainties related to scaling do not render our results/conclusions meaningless.

Regarding the emissivity used at the ground sites, excepting field visits and special cases where a wide array of sensors are available to account for the variability found at the site, everyone has to start from some assumption, because often we only have surface-leaving radiance to work with from the instruments. Wang and Liang 2009, for example, relied on MODIS Collection 5 broadband emissivity for their SURFRAD validations. We mention the use of CERES broadband emissivity (Wilber et al. 1999) for our SURFRAD validations, and CERES 11-µm emissivity (Chen et al. 2004) for our single-channel retrievals and ARM validations (page 4, lines 19 and 8, respectively). Specific emissivity values are provided in Table 2. Also, in a recent poster presented at the 2015 AGU (Scarino, B. R., P. Minnis, C. R. Yost, T. Chee, and R. Palikonda, 2015: Sensitivity of satellite-based skin temperature to different surface emissivity and NWP reanalysis sources demonstrated using a single-channel, viewing-angle-corrected retrieval algorithm. 2015 Fall Meeting American Geophysical Union, Dec 14-18 2015, San Francisco, CA.), we assessed the use of CERES broadband and 11-µm emissivity versus MODIS broadband and band 31 emissivity, respectively, for deriving LST, and found that the LSTs are comparable to within 0.01±0.16-K accuracy and 0.07±0.20-K precision relative to all SURFRAD and ARM validation sources (now stated on page 4, lines 19-21).

In addition to already acknowledging the potential bias introduced by the up-scaling problem on page 14, we explicitly mention the conclusion of Wang and Liang 2009 regarding

the comparison of point-like ground measurements with satellite-derived LST, and provide reasons why we believe it is important to continue reporting these non-scaled validation results, as described above (page 14, lines 24-28). Also, similar to Heidinger et al., we now caution users to be aware of this issue when using our data (page 14, line 29-30).

2. To first order the average view angle dependent correction is a move in the right direction but without dependence on elevation and surface type information it is hardly a complete correction, and may result in compensating errors dependent on pixel location.

The purpose of this paper is to introduce a simple, first-step empirical model that is based solely on VZA. Developing an all-encompassing model requires significant additional analysis and is more appropriate for a follow-on paper. Therefore, we stress that initial development must start regionally, and we highlight that this is a first step (page 8, lines 7-8). Although including more variables (e.g., relative azimuth, solar azimuth, land cover type, surface elevation, hemisphere, etc.) would certainly strengthen the model, such further subsetting with the present dataset would reduce sample size beyond reliability. As such, variable-based subsetting is more suitable for future improvements. Discussion of the need for this kind of subsetting has been added in Section 5.1 (page 9, lines 26-29), and is also reiterated in the Conclusions (page 16, lines 37-38).

3. Pinheiro et al (2006) and Guillevic et al (2013) have already shown that the nighttime LST is independent of the viewing considerations, so the relevancy of bias correction at night with this methodology is questionable. You are likely compensating for GOESMODIS time differences or spatial aggregation of MODIS LST.

We acknowledge the findings of Pinheiro et al. (2006) and Guillevic et al. (2013) in regards to nighttime LST VZA dependency on page 9, lines 20-22, where we also admit that our result confirming as such (Fig. 8) is unsurprising. Their analyses, however, were limited to a single surface type (woodland near Evora, Portugal or the Savannah) for a partial year, and do not provide universal conclusions. We do admit that using the daytime model at night may seem suspect, especially considering the results of Figs. 8 and 11 (see Section 5.4, page 13, lines 8-12). We can't ignore, however, that the GOES-East/West LST comparisons of Section 5.3, and the resultant Fig. 13, suggest that nighttime VZA dependency does matter. Furthermore, we cite Minnis and Khaiyer (2000) and Vinnikov et al. (2012), who note that nocturnal anisotropy can be induced by differential cooling between lower- and upper-level surfaces and different fractional amounts of vegetation in the sensor FOV (page 9, lines 17-19).

For now, we can only use the daytime model to assess nighttime validation, because we were unable to resolve a nighttime dependency with our standard methodology (again, see Fig. 8). We present the results of using the model for nighttime validation versus ground sites, which overall show that its application is beneficial. For example, relative to the ARM IRT, absolute daytime bias reduces from 1.39 K to 0.20 K, and absolute nighttime bias reduces from 0.95 K to 0.16 K. The SDD reduces in both cases as well. We'll reiterate that these are GEO and AVHRR nighttime ARM validation results, which are free from influence of any GOES-MODIS time difference or MODIS pixel aggregation (neither are a factor here). Taking all ground site comparison results into account, the average absolute bias reduction at night is 0.1 K, and the average SDD reduction is 2%. These overall nighttime improvements are now mentioned explicitly in the final paragraph of Section 5.4 (page 15, lines 23-24). Furthermore, we've added a sentence in the same paragraph, cautioning users that, although there is improvement, the nocturnal correction is nevertheless based on daytime

**observations (page 15, lines 24-26).**

4. There is no discussion or mention of the possible effects of aggregating MODIS 1-km to the effective GOES pixel resolution (assumed 4 km here?). Temperature does not scale up in a linear fashion (emissivity does), so you are introducing an additional uncertainty in your VZA corrections from the scaling.

**All pixels, whether from GOES, AVHRR, or MODIS (both our derived MODIS and MYD11), are aggregated as radiance values and then converted to mean LST. Specific discussion of this has been added to page 8, line 15, and page 10, lines 14-16. We recognize that temperature does not scale linearly with emissivity or radiance, but nevertheless as a matter of experimentation, we computed the non-linearity error: it would be at most 0.02-K if we had in fact aggregated LST values.**

5. Eq. 13. Without temperature/emissivity separation you can not simply imply an emissivity dependence from the surface leaving radiance fraction, which makes this section and all accompanying figures invalid assumptions.

**Logic follows that surface-leaving temperature as viewed from two different VZAs would be perceived as being different. That is, if the surface-leaving temperature at *VZA1* is *T1*, and at *VZA2* is *T2*, and if there is a VZA dependency, then *T2 ≠T1*. However, we know the actual temperature of the scene can only be one temperature, so the effective emissivity has to be the explanation for the difference. In other words, if the scene has only one true radiating temperature, then there must be some VZA-dependent factor that causes observed brightness temperatures to differ for different VZAs. Putting it very simply – because surface-leaving temperature can be constructed as a product of emissivity and the "true" skin temperature, then the VZA-dependent factor must be emissivity.**

Please double-check references (e.g. Wang et al. 2014) is not included.

**Thank you. All reference and citation oversights have been remedied.**